# RESTRAN: A GNN ALTERNATIVE TO LEARN A GRAPH WITH FEATURES

## ABSTRACT

This paper considers a vertex classification task where we are given a graph and associated vector features. The modern approach to this task is graph neural networks (GNNs). However, due to the nature of GNN architectures, GNNs are known to be biased to primarily learn homophilous information. To overcome this bias in GNN architectures, we take a simple alternative approach to GNNs. Our approach is to obtain a vector representation capturing both features and the graph topology. We then apply standard vector-based learning methods to this vector representation. For this approach, we propose a simple transformation of features, which we call *Resistance Transformation* (abbreviated as *ResTran*). We provide theoretical justifications for ResTran from the effective resistance, $k$-means, and spectral clustering points of view. We empirically demonstrate that ResTran is more robust to the homophilous bias than established GNN methods.

## 1 INTRODUCTION

This paper considers a vertex classification task in the "graph-with-features" setting. A dataset for this setting consists of a graph and a feature associated with each vertex. The task is to classify vertices using the features and the graph. The modern approach to this task is graph neural networks (GNNs) (Gori et al., 2005; Kipf & Welling, 2016a; Veličković et al., 2018). GNNs propagate features over the graph to build expressive latent embeddings; the embeddings are then consumed in downstream classification models. However, due to the nature of these GNN architectures, GNNs are typically known to have a bias towards homophilous information and to not be effective in learning heterophilous information (Hoang & Maehara, 2019; Luan et al., 2022). This bias worsens if we stack GNN layers (known as "over-smoothing" (Li et al., 2018; Oono & Suzuki, 2019)). Some recent GNN models mitigate this bias, such as (Azabou et al., 2023; Pei et al., 2020; Luan et al., 2021). However, such models, including these examples, often involve complicated GNN architectures.

In this paper, to overcome this homophilous bias in a simpler way, we propose an alternative approach to GNNs since this bias seems to be inherent in GNN architectures. Instead of mitigating biases by complicating the GNNs, our approach is to obtain a vector representation for the features and graph. Then, we apply standard vector-based learning methods to this vector representation, such as established neural network (NN) based models like variational autoencoder or even support vector machines (SVMs). For this approach, we propose a *Resistance Transformation* (abbreviated as *ResTran*), a simple transformation of feature vectors to incorporate graph structural information.

We theoretically justify ResTran from a connection between the $k$-means and spectral clustering. Our justification is inspired by (Dhillon et al., 2004), which justifies using feature maps for spectral clustering applied to vector data. For this purpose, Dhillon et al. (2004) takes the following steps as i) setting up $k$-means objective for transformed vectors by a feature map and ii) showing the equivalence from this $k$-means objective to spectral clustering. For ResTran, we follow a similar strategy: i) modifying the $k$-means to incorporate the vector representation by ResTran and ii) showing the equivalence from this $k$-means to spectral clustering. We show that this modified $k$-means for the featureless setting (i.e., looking only at a graph by taking features as an identity matrix) is equivalent to spectral clustering. Moreover, for the graph-with-features setting, we show that this $k$-means can be seen as a natural extension of spectral clustering from the featureless to the graph-with-features setting. We also discuss why ResTran may preserve the homophilous and heterophilous information better than the established GNNs. Our experiments show that ResTran outperforms graph-only and feature-only representation in unsupervised tasks. We also numerically show that ResTran is more robust to the homophilous bias than established GNNs in the semi-supervised learning (SSL) tasks.

**Contribution.** In summary, our contributions are as follows. i) We propose a simple *ResTran* for a graph-with-features problem. ii) We theoretically justify ResTran from an effective resistance, $k$-means, and spectral clustering perspective. iii) We numerically confirm that ResTran is more robust to homophilous bias than established GNNs for common datasets. *All proofs are in the Appendix.*

## 2 PRELIMINARIES

This section sets up the definition and reviews some foundations of graph learning.

### 2.1 GRAPH NOTATIONS AND VERTEX CLASSIFICATION PROBLEMS

**Graph Notations.** A graph $G = (V, E)$ is a pair of sets consisting of the *vertices $V$* and the *edges $E$*. Throughout this paper, we use $n := |V|$ and $m := |E|$. An edge connects two distinct vertices. We assume that our graph is *undirected*. We represent a graph by an *adjacency matrix* $A \in \mathbb{R}^{n \times n}$; the $ij$-th element and $ji$-th element of $A$ are the weight of the edge between $i$ and $j$, and $a_{ij} = a_{ji} := 0$ if there is no edge between $i$ and $j$. A *degree $d_i$* for a vertex $i$ is defined as $d_i := \sum_j a_{ij}$. We define the *degree matrix $D$*, a diagonal matrix whose diagonal elements are $D_{ii} := d_i$. We define the *graph Laplacian* as $L := D - A$ and a *normalized graph Laplacian* as $L_N := D^{+1/2} L D^{+1/2}$, where $^+$ is the pseudoinverse. Note that these Laplacians are positive semi-definite (PSD) matrices. We use $\mathbf{1} \in \mathbb{R}^n$ for the all one vector and $\mathbf{e}_i \in \mathbb{R}^n$ for the $i$-th coordinate vector. See (Bapat, 2010) for details.

**Graph-with-features Problem vs. Featureless Problem.** This paper considers a vertex classification task. This task is classifying vertices of the graph into $k$ classes. For this task, we consider two settings. *i) Graph-With-Features Problem.* This problem assumes that the $i$-th vertex is associated with $f$ dimensional *features* $\mathbf{x}_i \in \mathbb{R}^f$. We define a feature matrix as $X := (\mathbf{x}_1, \ldots, \mathbf{x}_n)$. A popular technique for this is a GNN. *ii) Featureless Problem.* This problem only considers the topology of the graph. There are various methods specifically for this, such as spectral clustering. We can also apply the graph-with-features methods to this featureless setting. A common technique to do so is by setting $X = I$, where $I$ is an identity matrix (Kipf & Welling, 2016a).

### 2.2 COORDINATE, LAPLACIAN COORDINATE, AND EFFECTIVE RESISTANCE

**Coordinate.** We define a *coordinate spanning set* induced from a symmetric PSD matrix $M$ as $\mathcal{V}_M := \{v_i := M^{+1/2} \mathbf{e}_i : i = 1, \ldots, n\}$. This coordinate spanning set is a coordinate in the discrete Hilbert space naturally defined for $M$. See Appendix C for details.

**Effective Resistance.** The effective resistance is an example where the coordinate spanning set by Laplacian (*Laplacian Coordinate*) plays a role. We consider an analog between a *connected* graph and an electric circuit; a vertex is a point at a circuit, and an edge is a resistor with resistance $1/a_{ij}$. A flow over a graph is mapped to a current, and $\mathbf{x} \in \mathbb{R}^n$ is seen as a potential at each vertex. We define an energy $S_G(\mathbf{x})$ for a potential $\mathbf{x}$ and effective resistance $r_G(i, j)$ between vertices $i$ and $j$ as

$$S_G(\mathbf{x}) := \mathbf{x}^\top L \mathbf{x}, \quad r_G(i, j) := (\min_{\mathbf{x}} \{S_G(\mathbf{x}) : x_i - x_j = 1\})^{-1}, \tag{1}$$

where $^\top$ is transpose. Using the Laplacian coordinate $\mathbf{v}_i, \mathbf{v}_j \in \mathcal{V}_L$, the $r_G(i, j)$ can be rewritten as

$$r_G(i, j) = \|\mathbf{v}_i - \mathbf{v}_j\|_2^2, \quad \text{where } \mathbf{v}_i, \mathbf{v}_j \in \mathcal{V}_L, \tag{2}$$

where $\| \cdot \|_2$ is the 2-norm. Recall that $\mathbf{v}_i = L^{+1/2} \mathbf{e}_i$ and $\mathbf{v}_j = L^{+1/2} \mathbf{e}_j$ by the definition above. In the following, we abbreviate effective resistance as *resistance*. Note that $r_G(i, j)$ can be seen as a distance between $\mathbf{v}_i, \mathbf{v}_j \in \mathcal{V}_L$. For more details, see (Doyle & Snell, 1984; Klein & Randić, 1993).

### 2.3 THE STANDARD $k$-MEANS AND WEIGHTED KERNEL $k$-MEANS

Since this paper is built on the $k$-means formulation, we briefly review this topic. Consider to partition the data points into $\{C_j\}_{j=1}^k$. The standard *k-means* is to minimize the following objective function.

$$\mathcal{J}(\{C_j\}_{j=1}^k) := \sum_{j \in [k]} \sum_{i \in C_j} \|\mathbf{x}_i - \mathbf{m}_j\|_2^2, \quad \mathbf{m}_j := \sum_{i \in C_j} \mathbf{x}_i / |C_j| \tag{3}$$

Minimizing $\mathcal{J}(\{C_j\}_{j=1}^k)$ is NP-hard (Mahajan et al., 2012). The approximated discrete solution is obtained by EM-type algorithms (Bishop, 2007). This $k$-means is generalized to the weighted and kernel setting. Let $\phi$ be a feature map. We define the *weighted kernel k-means* objective as

$$\mathcal{J}_\phi(\{C_j\}_{j=1}^k) := \sum_{j \in [k]} \sum_{i \in C_j} w(\mathbf{x}_i) \|\phi(\mathbf{x}_i) - \mathbf{m}_{\phi,j}\|_2^2, \mathbf{m}_{\phi,j} := \sum_{\ell \in C_j} w(\mathbf{x}_\ell) \phi(\mathbf{x}_\ell) / \sum_{\ell \in C_j} w(\mathbf{x}_\ell). \tag{4}$$

where $w(\mathbf{x}_i)$ is a weight at $\mathbf{x}_i$ and $\mathbf{m}_{\phi,j}$ serves as a weighted mean of the cluster $C_j$.

## 2.4 Graph Cut And Spectral Clustering

Consider partitioning a graph $G$ into two vertices sets $V_1 \cup V_2 = V$, $V_1 \cap V_2 = \varnothing$. For this partitioning, we define two objective functions to minimize, *normalized cut* and *ratio cut*, as

$$\mathrm{NCut}(V_1, V_2) := \sum_{i \in V_1, j \in V_2} a_{ij} \left( \frac{1}{\mathrm{vol}(V_1)} + \frac{1}{\mathrm{vol}(V_2)} \right), \mathrm{RCut}(V_1, V_2) := \sum_{i \in V_1, j \in V_2} a_{ij} \left( \frac{1}{|V_1|} + \frac{1}{|V_2|} \right),$$

where $\mathrm{vol}(V) := \sum_{i \in V} d_i$. We extend these to partition $V$ into $k$ subsets $V_i (i = 1, \ldots, k)$ where $V_i \cap V_j = \varnothing$ if $i \neq j$ and $\cup_{i=1}^{k} V_i = V$. For this $k$-way partitioning, we extend the cut objective functions as

$$\mathrm{kNCut}(\{V_i\}_{i=1}^k) := \sum_{i \in [k]} \mathrm{NCut}(V_i, V \backslash V_i), \mathrm{kRCut}(\{V_i\}_{i=1}^k) := \sum_{i \in [k]} \mathrm{RCut}(V_i, V \backslash V_i),$$

where $[k] := \{1, \ldots, k\}$. Minimizing these cut objectives is a discrete optimization and known as NP-hard (von Luxburg, 2007). Thus, we consider to relax these problems as follows. We introduce an indicator matrix $Z \in \{0, 1\}^{N \times k}$ and its variants for normalized cut $Z_N$ and for ratio cut $Z_R$ as

$$z_{ij} := \begin{cases} 1 & (i \in V_j) \\ 0 & (\text{otherwise}), \end{cases} \qquad Z_N := D^{1/2} Z (Z^\top D Z)^{-1/2}, \quad Z_R := Z(Z^\top Z)^{-1/2}. \tag{5}$$

Note that $Z_N^\top Z_N = I$ and $Z_R^\top Z_R = I$. Note also that $(z_N)_{ij} = \sqrt{d_i / \mathrm{vol}(V_j)}$ and $(z_R)_{ij} = \sqrt{1/|V_j|}$ if $i \in V_j$ otherwise 0. Using these indicator matrices, we can see the following.

**Proposition 1** (classical, e.g., Yu & Shi (2003)). *Ratio and normalized cuts are rewritten as follows.*

$$\min \mathrm{kNCut}(\{V_i\}_{i=1}^k) = \min_{Z_N} \{\mathrm{trace}(Z_N^\top L_N Z_N) \text{ s.t. } Z_N^\top Z_N = I\} \tag{6}$$

$$\min \mathrm{kRCut}(\{V_i\}_{i=1}^k) = \min_{Z_R} \{\mathrm{trace}(Z_R^\top L Z_R) \text{ s.t. } Z_R^\top Z_R = I\}, \tag{7}$$

*Moreover, let $\lambda_i$ and $\lambda_{N,i}$ be the $i$-th eigenvalues $L$ and $L_N$ respectively. Relaxing $Z_R$ and $Z_N$ into real values, we have $\min \mathrm{kNCut}(\{V_i\}_{i=1}^k) = \sum_{i=1}^k \lambda_{N,i}$ and $\min \mathrm{kRCut}(\{V_i\}_{i=1}^k) = \sum_{i=1}^k \lambda_i$*

From Prop. 1, relaxing $Z_N$ and $Z_R$ into real values, minimizing cut objectives become eigenproblems of $L_N$ and $L$, which is not NP-hard. Solving these eigenproblems is called *spectral clustering*.

## 2.5 Spectral Connection: Weighted Kernel $k$-means to Spectral Clustering

The spectral clustering discussed above is applied to a given graph. By using feature maps, we may apply spectral clustering to given vector data. (Dhillon et al., 2004) provides one justification for the use of feature maps are through the connection between $k$-means and spectral clustering viewpoint. Since our work is inspired by this justification, we review this topic.

We assume vector data $X = (\mathbf{x}_1, \ldots, \mathbf{x}_n)$. A common practice to apply spectral clustering to $X$ is to use a feature map. Namely, we apply spectral clustering on a graph obtained as $a_{ij} := \langle \phi(\mathbf{x}_i), \phi(\mathbf{x}_j) \rangle$. Note that we choose $\phi$ so that $a_{ij} \geq 0$, for all $i, j \in [n]$. The question is, *how may the use of feature maps be justified?* Justification can be done by many ways. We review an established justification by (Dhillon et al., 2004), which develops the following strategy.

1. Using vectors transformed by a feature map $\phi(\mathbf{x}_i)$ to the weighted kernel $k$-means.
2. Showing a connection from this weighted kernel $k$-means to the spectral clustering.

By this, we can ground the use of a feature map to spectral clustering through $k$-means lens. (Dhillon et al., 2004) shows the following claim.

**Proposition 2** ((Dhillon et al., 2004)). *Informal. Consider a graph $a_{ij} = \langle \phi(\mathbf{x}_i), \phi(\mathbf{x}_j) \rangle$ and its degree $d_i$. We apply spectral clustering to this graph $A$. We substitute a weight $w(\mathbf{x}_i) = 1/d_i$ to the weighted kernel $k$-means $\mathcal{J}_\phi(\{V_j\}_{j=1}^k)$ Eq. (4). Then, in a "relaxed sense," we obtain*

$$\min \mathcal{J}_\phi(\{V_j\}_{j=1}^k) = \min \mathrm{kNCut}(\{V_i\}_{i=1}^k) \tag{8}$$

By this connection, we may say that the spectral clustering for vector data via kernel is justified from a weighted kernel $k$-means view. Note that we observe this connection only for the normalized cut. For more details including the formal statement of Prop. 2, see (Dhillon et al., 2004) and Appendix G.

## 2.6 Homophily, Heterophily, and Eigenspace of Laplacian

A graph dataset may be classified into two notions. The *homophily* assumption is that adjacent vertices are more likely to be in the same group. The *heterophily* assumption is that vertices are collected in

---

**Algorithm 1** Proposed Practical Framework for SSL via ResTran and Krylov Subspace Method

---

**Input:** Graph $G = (V, E)$, Features $X$, Training and Test Indices $Tr, Te$, Krylov Subspace Dim $r$

Obtain the approximated ResTran $\tilde{X}_G$ (Eq. (10)) by applying Krylov subspace method, i.e.,
$$\tilde{X}_G = \text{KRYLOVSUBSPACEMETHOD}(L, X, r)$$
Obtain the model by applying any vector ML method to the training data whose indices are $Tr$ as
$$\text{MODEL} = \text{ANYVECTORMLMETHOD}(\{(\tilde{X}_G)_{\cdot i}, y_i\}_{i \in Tr})$$
Obtain the predicted label $\hat{\mathbf{y}}$ by applying MODEL to the test data whose indices are $Te$ as
$$\hat{\mathbf{y}} = \text{MODEL}(\{(\tilde{X}_G)_{\cdot i}\}_{i \in Te})$$

**Output:** The predicted label $\hat{\mathbf{y}}$

---

diverse groups, i.e., the contrary to homophily assumption. From the cut definition, spectral clustering assumes homophily. Recall that the spectral clustering looks at the eigenspace associated with smaller eigenvalues (i.e., low-frequencies) of $L$. Thus, we may see that this eigenspace contains homophilous information. Also, we may say that the eigenspace for larger eigenvalues (i.e., high-frequencies) of $L$ captures heterophilous information. In the following, we say "low-frequency" for homophily or "high-frequency" for heterophily. See (Hoang & Maehara, 2019; Luan et al., 2022) for details.

## 3 PROPOSED METHOD: RESTRAN

This section presents our learning framework for the graph-with-features setting. A common method for this setting is a GNN, where we develop NNs incorporating a graph. Instead, we propose a vector representation of the graph-with-features, which we call *ResTran*. We then apply vector-based ML methods to this vector, e.g., SVM and the standard NN methods. In Sec. 4, we will justify ResTran from the spectral connection and resistance view and also explore characteristics of ResTran.

For our framework, we use the *shifted graph Laplacian*, as done in (Herbster & Pontil, 2006), as

$$L_b^{-1} := L^+ + bJ_G, \quad \text{where} \quad b > 0, (J_G)_{ij} := \begin{cases} 1 & (i \text{ and } j \text{ are in the same component}) \\ 0 & (\text{otherwise}), \end{cases} \tag{9}$$

Note that from the definition $J_G = \mathbf{1}\mathbf{1}^\top$ if the graph is connected, i.e., contains only one component. Note also that $L_b$ is invertible since $L_b$ is symmetric positive definite (PD) as we see later in Prop. 3.

**Proposed Framework via *ResTran*.** Below we propose our framework. The overall strategy is to i) have a vector representation of graph-with-features ii) apply a vector based ML method. For i), using the coordinate $\mathcal{V}_{L_b}$, we propose our *Resistance Transformation* (*ResTran* for abbreviation) $X_G$ as

$$X_G := (\mathbf{x}_{G,1}, \ldots, \mathbf{x}_{G,n}), \quad \text{where } \mathbf{x}_{G,i} := X\mathbf{v}_i', \ \mathbf{v}_i' \in \mathcal{V}_{L_b}. \tag{10}$$

Recall that $\mathbf{v}_i' = L_b^{-1/2}\mathbf{e}_i$ by definition of $\mathcal{V}_{L_b}$ in Sec. 2.2. Note that $\mathbf{x}_i, \mathbf{x}_{G,i} \in \mathbb{R}^f$ and $X, X_G \in \mathbb{R}^{n \times f}$. For ii), we then use any vector based ML methods for $X_G$, such as SVM and NN-based methods.

**Practical Implementation via Krylov Subspace Method.** If we naively compute $L_b^{-1/2}$ and then multiply $X$ to obtain ResTran Eq. (10), it costs prohibitive $O(n^3)$ complexity due to the computation of $L_b^{-1/2}$. Instead of this naive computing, we consider to approximate $X_G$. For this purpose, we apply the Krylov subspace method, by which we can approximate a solution of linear algebraic problems. The Krylov subspace method reduces the computational complexity from $O(n^3)$ to $O(rfm)$, where $r$ is the dimension of the Krylov subspace. The dimension $r$ is typically small, say $r < 100$. The Krylov subspace method approximates $X_G$ by considering $L$ and $X$ *at the same time*. Thus, we expect a better approximation for Krylov than approximating $L_b^{-1/2}$ without using $X$; such methods include the polynomial approximation. Note that this polynomial approximation is common in the established convolutional GNNs, such as (Defferrard et al., 2016; Kipf & Welling, 2016a). Refer to Appendix A or (Higham, 2008) for details. The overall proposed framework is summarized in Alg. 1. Note that Alg. 1 can be interpreted as SSL even though we apply supervised methods such as SVM because we first observe $X$ and $G$ to obtain $X_G$. This is same as GNNs, where we observe $X$ and $G$ before we learn. Alg. 1 naturally generalizes to the unsupervised setting.

**Coordinate Interpretation of ResTran.** We first remark that $L_b^{-1/2} = (\mathbf{v}_1', \ldots, \mathbf{v}_n')$, $L_b^{-1/2}$ is symmetric, and $X_G = XL_b^{-1/2}$. The $X_G^\top$ can be seen as retaking basis of $X^\top$ by $\mathcal{V}_{L_b}$ if we see $L_b^{-1/2}$ in row-wise. Moreover, by comparing the original $X = (X\mathbf{e}_1, \ldots, X\mathbf{e}_n)$, the $X_G$ can be seen as retaking $\mathbf{e}_i$ to $\mathbf{v}_i'$ to indicate $i$-th vertex if we see $L_b^{-1/2}$ in column-wise.

**Comparison with GNNs.** This approach is simpler than existing GNN approaches. The recent GNNs often involve complicated graph designs in layers of NN or pre/post-processing. However, our framework is simple since we transform $X$ to $X_G$ and then apply any vector-based methods.

## 4    CHARACTERISTICS AND JUSTIFICATION OF RESTRAN

This section discusses the characteristics and justification of ResTran. We first discuss the characteristics of ResTran, by exploring theoretical properties of Laplacian coordinate $\mathcal{V}_{L_b}$ from a resistance view. Next, we justify using ResTran of $X_G$ from a $k$-means perspective.

### 4.1    CHARACTERISTICS OF RESTRAN: AN EFFECTIVE RESISTANCE VIEW

This section discusses the characteristics of ResTran. We first explore theoretical properties of the Laplacian coordinate $\mathcal{V}_{L_b}$. We then interpret these results to explain characteristics of ResTran.

**Theoretical Properties of $\mathcal{V}_{L_b}$.** In the following, we assume that we have $K$ connected components. We write $G_i := (V_i, E_i)$ for $i = 1, \ldots, K$, and $G = G_1 \cup \ldots \cup G_K$. We write as $n_i := |V_i|$. Whiteout loss of generality, we can assume that $n_1 \leq \ldots \leq n_K$. Denote $\mathbf{1}_{G_j}$ by all one vector for $G_j$, i.e., $(\mathbf{1}_{G_j})_i = 1$ if $j \in V_{G_j}$ otherwise 0. Note that $\sum_{j \in [K]} \mathbf{1}_{G_j} = \mathbf{1}$ and $(J_G)_i = \mathbf{1}_{G_s}$ if $i \in V_s$. Note also that $\mathbf{1}_{G_j}$ are eigenvectors of $L$. Using this notation, we have properties of $\mathcal{V}_{L_b}$ as follows.

**Proposition 3.** *Suppose that a graph $G$ has $K$ connected components. Let $(\lambda_i, \boldsymbol{\psi}_i)$ be the $i$-th eigenpair of $L$. If $n_1 b > \lambda_{K+1}^{-1}$, the $i$-th eigenpair $(\lambda_i', \boldsymbol{\psi}_i')$ of $L_b^{-1/2}$ is*

$$(\lambda_i', \boldsymbol{\psi}_i') = \begin{cases} \left(\lambda_{n+1-i}^{-1/2}, \boldsymbol{\psi}_{n+1-i}\right) & \text{for } i = 1, \ldots, n - K, \\ \left((n_{i-(n-K)}b)^{1/2}, \mathbf{1}_{G_{n_{i-(n-K)}}}\right) & \text{for } i = n - K + 1, \ldots, n. \end{cases}$$

**Corollary 4.** $L_b^{-1/2}\mathbf{e}_i = (L^{+1/2} + \sqrt{b}J_G^{1/2})\mathbf{e}_i$, *where* $J_G^{1/2} = \sum_i^K (n_i^{-1/2}\mathbf{1}_{G_i}\mathbf{1}_{G_i}^\top)$

This proposition shows that $L$ and $L_b^{-1/2}$ share eigenvectors and that $L_b^{-1/2}$ is PD since $\lambda_i' > 0$ for all $i$. Next, we explore the characteristics of the coordinates $\mathcal{V}_{L_b}$. We define an *extended resistance* as

$$r_{G,b}'(i,j) := \|\mathbf{v}_i' - \mathbf{v}_j'\|_2^2, \quad \mathbf{v}_i', \mathbf{v}_j' \in \mathcal{V}_{L_b} \tag{11}$$

Recall that $\mathbf{v}_i' = L_b^{-1/2}\mathbf{e}_i$. The following can be claimed.

**Proposition 5.** *If two vertices $i, j$ in the same component $G_s$, $r_{G,b}'(i,j) = r_{G_s}(i,j)$.*

Prop. 5 means that even if we use $\mathcal{V}_{L_b}$ instead of $\mathcal{V}_L$, the resistance, the distance between coordinates (Eq. (2)), is preserved within the connected component. For inter-component, the parameter $b$ controls the connectivity among the components. If two vertices are in different components, it is natural to think that they are apart. However, in the graph-with-features setting, even if two vertices are in different components, the two vertices often belong to the same cluster; therefore, these are not apart so much. We parameterize this intuition by $b$; by taking larger $b$, we weigh more on the disconnected observation. Taking $b$ large enough for two vertices $i, \ell$ in the different components, we can make $r_{G,b}'(i,\ell)$ greater than *any* resistances within the component as follows.

**Proposition 6.** *If $b > \sqrt{2}n_1/\lambda_{K+1}$, $r_{G,b}'(i,\ell) > r_{G,b}'(i,j)$ for $i, j \in V_s$ and $\ell \in V_t$ where $s \neq t$.*

Using these theoretical properties, we observe the following characteristics of the ResTran.

**ResTran from a Resistance View.** From Prop. 5 and Prop. 6, we observe that $\mathcal{V}_{L_b}$ serves as a coordinate offering an extended resistance. Our ResTran may be viewed as the basis transformation from $\mathbf{e}_i$ to $\mathbf{v}_i'$. This is why we call our transformation Eq. (10) as a "resistance" transformation.

**ResTran Capturing a Mix of Homophilous and Heterophilous Information.** Our ResTran can be seen as favoring the homophilous assumption but, at the same time, not ignoring the heterophilous assumption, while GNNs are biased toward homophily. Recall that the homophilous information is contained in the space spanned by $\psi_i$ for the smaller eigenvalues $\lambda_i$ while the heterophilous information is in the space spanned by $\psi_j$ for larger eigenvalues $\lambda_j$, as seen in Sec. 2.6. GNNs are effective at homophilous data but not at heterophilous data (Luan et al., 2022). Loosely speaking, this happens because each layer of GNNs multiplies the adjacency matrix $A$ to the next layer, often several times (see Appendix B for details). Stacking the layers enlarges the low-frequency components,

which leads to a bias towards homophily. On the other hand, ResTran "balances" homophily and heterophily. Observe that we can see that $L_b^{-1/2}$ is "spectral reordering" of the graph Laplacian $L$ (see Prop. 3); the largest eigenvalues of $L_b^{-1/2}$ are the smallest eigenvalues of $L$, and the order is reversed. Also, from Prop .3, the eigenvalues of $L_b^{-1/2}$ associated with eigenvectors $\psi_i$ is either $\sqrt{n_i b}$ or $\lambda_i^{-1/2}$, which is large since $\lambda_i$ is small. Recall that ResTran multiplies $L_b^{-1/2}$ to $X$ once. Thus, the space containing the homophilous information is amplified by large $\lambda_i^{-1/2}$. At the same time, we do not ignore the heterophilous space, but this is amplified by small $\lambda_j^{-1/2}$ since $\lambda_j$ is large.

## 4.2 JUSTIFICATION OF RESTRAN $X_G$ FROM A $k$-MEANS PERSPECTIVE

This section justifies our ResTran $X_G$. Our justification is inspired by (Dhillon et al., 2004). As reviewed in Sec. 2.5, Dhillon et al. (2004) justifies using a feature map for spectral clustering applied to vector data. For this purpose, Dhillon et al. (2004) use the following steps: i) modify the $k$-mean objectives to incorporate a vector transformed by a feature map and ii) show a connection from this modified $k$-means objective to spectral clustering. Here, we aim to establish a similar connection for ResTran. For this purpose, following i), we use $X_G$ in the $k$-means objective Eq. (3) as

$$\mathcal{J}_G(\{V_i\}_{i=1}^k) := \sum_{j \in [k]} \sum_{i \in V_j} \|\mathbf{x}_{G,i} - \mathbf{m}_{G,j}\|_2^2, \quad \mathbf{m}_{G,j} := \sum_{\ell \in V_j} \mathbf{x}_{G,\ell}/|V_j|. \tag{12}$$

This objective is a replacement of the standard $k$-means Eq. (3) from $\mathbf{x}_i$ to $\mathbf{x}_{G,i}$. Following ii), we establish connections from this $k$-means objective to spectral clustering as follows.

- Sec. 4.2.1 shows that in the featureless setting where $X = I$, conducting $k$-means on $\mathbf{v}_i' = L_b^{-1/2}\mathbf{e}_i$ is equivalent to spectral clustering.
- Sec. 4.2.2 shows that conducting $k$-means on $\mathbf{x}_{G,i}$ can be seen as a natural generalization of the spectral clustering through the $k$-means discussion.

With these connections, we say that ResTran is justified in the same sense as the feature map for spectral clustering as done by (Dhillon et al., 2004) discussed in Sec. 2.5.

### 4.2.1 JUSTIFICATION FOR FEATURELESS SETTING: REVISITING THE SPECTRAL CONNECTION

This section justifies Eq. (12) for the featureless setting, where we use $X = I$. Therefore, for featureless setting, $X_G = (\mathbf{v}_1', \ldots, \mathbf{v}_n')$ from the definition of $X_G$ Eq. (10). Using this $X_G$, we can rewrite Eq. (12) and further expand using Frobenius norm $\|\cdot\|_{\mathrm{Fro}}$ and indicator matrix $Z_R$ (Eq. (5)) as

$$\mathcal{J}_R(\{V_j\}_{j=1}^k) := \sum_{j \in [k]} \sum_{i \in V_j} \|\mathbf{v}_i' - \mathbf{m}_j\|_2^2, \quad \mathbf{m}_j := \sum_{i \in V_j} \mathbf{v}_i'/|V_j|, \mathbf{v}_i' \in \mathcal{V}_{L_b} \tag{13}$$

$$= \|L_b^{-1/2} - Z_R Z_R^\top L_b^{-1/2}\|_{\mathrm{Fro}}^2. \quad (\because \mathbf{m}_j = (L_b^{-1/2} Z_R Z_R^\top)._i \text{ if } i \in C_j). \tag{14}$$

With Eq. (14), we may obtain the *relaxed* solution of $k$-means by relaxing $Z_R$ into real values. We first claim that the objective Eq. (13) grounds on the extended resistance (Eq. (11)) as follows.

**Proposition 7.**

$$\mathcal{J}_R(\{V_j\}_{j=1}^k) = \sum_{j \in [k]} \sum_{i,\ell \in V_j} r_{G,b}'(i,\ell)/|V_j| \tag{15}$$

This proposition means that the $k$-means objective using $\mathbf{v}_i'$ (Eq. (13)) can be seen as the sum of the extended resistances. Since Eq. (15) itself seems a natural objective for graph clustering, our $k$-means Eq. (13) also may be seen as a natural objective. We also show that minimizing $\mathcal{J}_R(\{V_j\}_{j=1}^k)$ (Eq. (13) and its equivalence Eq. (15)) has a theoretical connection to spectral clustering as follows;

**Theorem 8.** *If we relax $Z_R$ into real values and $n_1 b > \lambda_{K+1}^{-1}$, we have*

$$\arg\min_{Z_R}\{\mathrm{RCut}(\{V_j\}_{j=1}^k) \text{ s.t. } Z_R^\top Z_R = I\} = \arg\min_{Z_R}\{\mathcal{J}_R(\{V_j\}_{j=1}^k) \text{ s.t. } Z_R^\top Z_R = I\} \tag{16}$$

This theorem means that that ratio cut and $k$-means using $\mathbf{v}_i'$ are theoretically equivalent if we relax $Z_R$. By this theorem, Eq. (13), featureless version of Eq. (12) using the common featureless technique $X = I$, are theoretically justified in a sense of $k$-means.

Remark that Thm. 8 revisits the spectral connection between $k$-means and spectral clustering as seen in Sec. 2.5. However, the previous connections only hold for the vector data and a feature map, not

for the discrete graph data like Thm. 8. Moreover, from Prop. 7 and Thm. 8, the clustering using resistance and spectral clustering are equivalent in a relaxed sense, which the previous connections have not shown. Finally, while the previous connections only hold for the normalized cut, Thm. 8 is the first to show the spectral connection for the ratio cut. Note that Thm. 8 naturally generalizes to normalized cut. For more details on how the previous connection and Thm. 8 differ, see Appendix G.

### 4.2.2 JUSTIFICATION FOR THE GRAPH-WITH-FEATURES SETTING: A $k$-MEANS VIEW

This section justifies the $k$-means objective for the graph-with-features setting Eq. (12). In Sec. 4.2.1, we saw that Eq. (13), which is a featureless setting of Eq.(12), is equivalent to the spectral clustering. This section shows that Eq. (12) is a "natural extension" of spectral clustering through Eq.(13).

We first recall that the common technique (see, e.g., Kipf & Welling (2016a;b)) to apply a graph-with-features method to featureless setting is substituting $X = I$. Thus, it is natural to think in a "reverse way"; in order to generalize the featureless methods to graph with the features method, we replace $I$ to the feature vector $X$. Since Eq. (14) is for a featureless setting, we now explicitly write $I$ as

$$\mathcal{J}_R(\{V_i\}_{i=1}^k) = \|L_b^{-1/2}I - Z_R Z_R^\top L_b^{-1/2}I\|_{\text{Fro}}^2. \tag{17}$$

Looking at Eq. (17), this can be thought as a featureless setting of the following objective function;

$$\mathcal{J}_G'(\{V_i\}_{i=1}^k) := \|L_b^{-1/2}X^\top - Z_R Z_R^\top L_b^{-1/2}X^\top\|_{\text{Fro}}^2. \tag{18}$$

Using $\mathbf{m}_{G,j}$ in Eq. (12), we further rewrite Eq. (18) as

$$\mathcal{J}_G'(\{V_i\}_{i=1}^k) = \sum_{j\in[k]}\sum_{i\in V_j}\|\mathbf{x}_{G,i} - \mathbf{m}_{G,j}\|_2^2 = \mathcal{J}_G(\{V_i\}_{i=1}^k), \tag{19}$$

by which we show that $\mathcal{J}_G(\{V_i\}_{i=1}^k)$ Eq. (12) and $\mathcal{J}_G'(\{V_i\}_{i=1}^k)$ Eq.(18) are equal.

What does the equivalence between $\mathcal{J}_G(\{V_i\}_{i=1}^k)$ and $\mathcal{J}_G'(\{V_i\}_{i=1}^k)$ mean? We begin with $\mathcal{J}_G'(\{V_i\}_{i=1}^k)$. The objective $\mathcal{J}_G'(\{V_i\}_{i=1}^k)$ can be seen as a generalization of $\mathcal{J}_R(\{V_i\}_{i=1}^k)$ (Eq.(13)) from featureless to graph-with-features setting. Recall that from Thm. 8, the featureless $\mathcal{J}_R(\{V_i\}_{i=1}^k)$ is equivalent to the standard spectral clustering. Thus, by stretching this idea from the featureless to the graph-with-features, $\mathcal{J}_G'(\{V_i\}_{i=1}^k)$ can be seen as a natural extension of spectral clustering to graph-with-features setting through a $k$-means perspective. Hence, since $\mathcal{J}_G'(\{V_i\}_{i=1}^k) = \mathcal{J}_G(\{V_i\}_{i=1}^k)$, we may say that the $k$-means $\mathcal{J}_G(\{V_i\}_{i=1}^k)$ we initially discuss in Eq. (12) can be seen as a natural "extended" spectral clustering for graph-with-features, seen through a $k$-means lens. Thus, we now establish a connection from $k$-means to the "extended" spectral connection using the common technique from the featureless to graph-with-features. In this sense, we may justify using $X_G$, similarly to (Dhillon et al., 2004). See Appendix I for more formulation.

Finally, Thm. 8 also offers insights into the graph-with-features setting. From Thm. 8, we see that the basis $\mathbf{v}_i'$ has a graph structural information through spectral clustering. Thus, we can say that the ResTran $x_{G,i}$ captures more graph structure than $\mathbf{x}_i$ since ResTran replaces the basis from $\mathbf{e}_i$ to $\mathbf{v}_i'$.

## 5 RELATED WORK

This section provides the review of the related work to our ResTran.

**Spectral Connection.** Our justification relies on the connection between spectral clustering, resistance, and $k$-means. The spectral clustering using ratio and normalized cut has been extensively studied (von Luxburg, 2007). The spectral connection for the normalized cut has been developed, such as (Bach & Jordan, 2003; Dhillon et al., 2004; Saito, 2022). However, the connection between ratio cut, resistance, and $k$-means are only loosely studied (Saerens et al., 2004; Zha et al., 2001), while Prop. 7 and Thm. 8 offer the exact connection. Also, the previous studies do not offer the interpretation as discussed in Sec. 4.2. We discuss more details in Appendices B and G.

**GNNs and SSL.** Our ResTran considers the graph-with-features setting. One popular approach to this task is GNN. GNNs are firstly proposed as NNs applied to the graph structural data (Gori et al., 2005; Scarselli et al., 2008). The GCN (Kipf & Welling, 2016a) and GAT (Veličković et al., 2018) are established methods. The recent advancements include (Hamilton et al., 2017; Wu et al., 2019; Alfke & Stoll, 2021) to name a few; see (Wu et al., 2020) for a survey. Moreover, Transformers using the spectral properties are considered (Dwivedi et al., 2023; Wang et al., 2022). While these GNNs

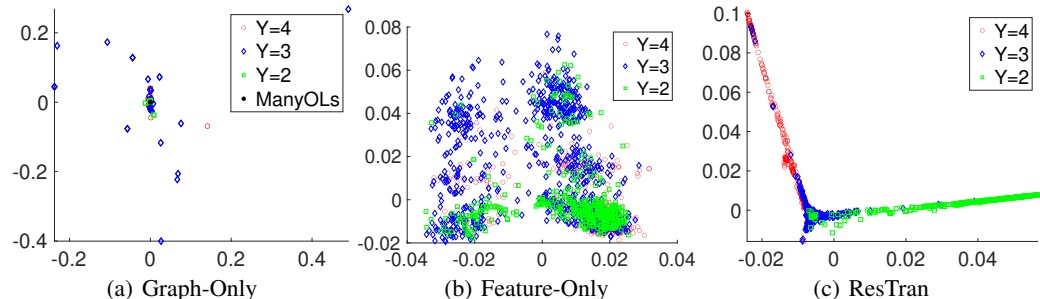

(a) Graph-Only          (b) Feature-Only          (c) ResTran

Figure 1: Plots of the eigenvectors of the graph Laplacian $L$ for three settings of Cora. For graph-only, we directly use a graph. For feature-only and ResTran, we compose a graph by applying a Gaussian kernel. We observe many overlaps of the points at the center, shown as "Many OLs."

Table 1: Experimental results for unsupervised learning. All measures are accuracy (%). "Graph-Only" uses only graph Laplacian. "Feature-only" uses a Gram matrix constructed only by features. "Graph + Feature" uses a Gram matrix constructed by our proposal $X_G$.

|  | Cora | Citeseer | Pubmed | Texas | Cornell | Wisconsin |
|---|---|---|---|---|---|---|
| Graph-Only | $29.3 \pm 0.5$ | $23.7 \pm 0.0$ | $39.6 \pm 0.0$ | $49.6 \pm 1.1$ | $49.0 \pm 5.6$ | $45.6 \pm 4.2$ |
| Feature-Only | $32.6 \pm 0.6$ | $45.5 \pm 0.9$ | $45.4 \pm 0.0$ | $55.2 \pm 0.5$ | $55.2 \pm 0.0$ | $47.8 \pm 0.0$ |
| Graph + Feature (Ours) | $\mathbf{58.9 \pm 4.5}$ | $\mathbf{48.2 \pm 0.8}$ | $\mathbf{71.6 \pm 0.6}$ | $\mathbf{55.5 \pm 0.5}$ | $\mathbf{55.7 \pm 0.0}$ | $\mathbf{48.2 \pm 0.3}$ |

need some specific design incorporating a graph structure into NNs, ours simply can apply existing vector learning methods to ResTran. The most relevant problem to this study is the homophily bias of GNNs, where GNNs have a bias to primarily learn homophilous information (Hoang & Maehara, 2019; Zheng et al., 2022; Luan et al., 2022; Platonov et al., 2023). This phenomenon worsens if we stack GNN layers, known as "over-smoothing" (Li et al., 2018; Oono & Suzuki, 2019). By construction, ResTran is expected to represent not only homophilous but also heterophilous information as discussed in Sec. 4.1. Finally, since most of the GNN studies are evaluated on SSL tasks, we mention a survey of SSL (Van Engelen & Hoos, 2020). For more details, see Appendices B.

## 6 EXPERIMENTS

This section numerically demonstrates the performance of ResTran. The purpose of our experiments is to evaluate if our ResTran $X_G$ improves i) the graph-only or feature-only representation and ii) the existing GNN methods. Recall that we propose to use ResTran $X_G$ and to apply a vector-based ML method. Thus, various sophistication can be involved for both ResTran and the comparison methods. However, to focus on evaluating our ResTran, we want to exclude the effects of sophistication as much as possible. To do so, our experiments only used simple and established methods for both ResTran and the comparison. We used Alg. 1 for ResTran. We evaluated ResTran and existing methods by accuracy, same as the previous studies such as (Kipf & Welling, 2016a). We evaluated ResTran on the standard homophilous and heterophilous datasets for the graph-with-features task. For the homophilous datasets, we used citation network datasets (Cora, Citeceer, and Pubmed) and purchase datasets (Amazon Photo and Computer). For heterophilous datasets, we used web datasets (Wisconsin, Cornell, Texas, Chameleon, Squirrel, and Actor). The details of the experiments, such as choices of ML models, hyperparameters, and architectures of the NNs, are described in Appendix H.

### 6.1 COMPARING RESTRAN WITH GRAPH-ONLY AND FEATURE-ONLY

This experiment briefly evaluates if our ResTran for representing the graph-with-features datasets improves the feature-only $X$ and graph-only $A$. If we observe that the latent space is more separable for ResTran $X_G$ than for graph-only and feature-only settings, we can say that ours improves the representation. For this purpose, we compare these on the simple setting, spectral clustering. For the feature-only and ResTran, we used the Gaussian kernel to form a graph and applied spectral clustering. For graph-only, we used the graph Laplacian for the spectral clustering. We conducted a simple $k$-means on the first $k$ eigenvectors of the graph Laplacian, and we reported the average. We first observe that Fig. 1 shows the plots of the second and third eigenvectors of the graph Laplacians for graph-only, feature-only, and ResTran of Cora. We see that ResTran offers better separation than graph-only and feature-only. The results of the unsupervised task are summarized in Table 1. In

Table 2: Experimental results for homophilous data using semi-supervised learning with some known labels. We use 5% labels. All measures are accuracy (%).

|  | Type | cora | citeseer | pubmed | photo | computer |
|---|---|---|---|---|---|---|
| GCN | GNN | **79.9 ± 0.9** | 67.4 ± 1.1 | 83.8 ± 0.4 | 83.1 ± 1.2 | 80.4 ± 0.4 |
| GAT | GNN | 74.9 ± 4.2 | 67.6 ± 0.1 | 82.8 ± 0.2 | **87.7 ± 1.3** | 80.3 ± 1.2 |
| SGC | GNN | 79.3 ± 1.7 | 70.2 ± 0.8 | 67.9 ± 1.8 | 80.1 ± 2.9 | 81.4 ± 2.0 |
| ResTran + LP | Non-NN | 30.6 ± 0.6 | 20.6 ± 4.6 | 39.5 ± 1.4 | 25.3 ± 0.2 | 37.5 ± 2.2 |
| ResTran + SVM | Non-NN | 49.1 ± 5.7 | 45.5 ± 6.7 | 76.5 ± 2.2 | 24.3 ± 2.7 | 43.8 ± 3.4 |
| ResTran + VAT | NN | 77.6 ± 2.5 | 68.7 ± 1.1 | 82.8 ± 0.7 | 86.3 ± 0.8 | 78.1 ± 2.4 |
| ResTran + AVAE | NN | 78.2 ± 1.8 | **71.7 ± 1.0** | **83.9 ± 0.7** | 86.8 ± 1.5 | **81.6 ± 0.9** |

Table 3: Experimental results for heterophilous data using semi-supervised learning with some known labels. We use 5% labels. All measures are accuracy (%).

|  | Type | Texas | Cornell | Wisconsin | chameleon | squirrel | actor |
|---|---|---|---|---|---|---|---|
| GCN | GNN | 50.9 ± 4.2 | 37.4 ± 9.3 | 46.3 ± 4.9 | 32.7 ± 2.0 | 23.5 ± 1.1 | 25.9 ± 0.9 |
| GAT | GNN | 50.3 ± 3.3 | 44.9 ± 4.9 | 44.0 ± 4.8 | 32.8 ± 1.8 | 23.4 ± 1.3 | 26.4 ± 0.9 |
| SGC | GNN | 44.6 ± 5.0 | 42.3 ± 5.3 | 44.6 ± 5.0 | 31.8 ± 1.8 | 23.5 ± 0.8 | 26.0 ± 0.8 |
| ResTran + LP | Non-NN | 46.3 ± 17.3 | 42.2 ± 20.6 | 37.3 ± 12.6 | 20.3 ± 0.8 | 20.0 ± 0.3 | 22.3 ± 2.8 |
| ResTran + SVM | Non-NN | 48.8 ± 14.1 | 45.7 ± 16.8 | 47.8 ± 9.6 | 33.6 ± 5.8 | 31.9 ± 0.9 | 29.4 ± 0.9 |
| ResTran + VAT | NN | **55.9 ± 5.1** | **49.0 ± 3.8** | **51.2 ± 5.0** | 34.0 ± 1.4 | 27.7 ± 3.5 | 27.8 ± 1.2 |
| ResTran + AVAE | NN | 51.4 ± 3.7 | 48.2 ± 3.7 | 50.0 ± 2.1 | **40.7 ± 1.4** | **32.4 ± 0.8** | **29.5 ± 1.3** |

all datasets, we see that ResTran improves both graph-only and feature-only. These results further confirm that ResTran $X_G$ better represents the dataset than the feature only $X$ or the graph only $A$.

## 6.2 Comparing ResTran with GNN Methods.

This section numerically evaluates if ResTran improves the existing GNN methods. We evaluate this on the SSL tasks. For comparison, we used three established simple GNN models, GCN (Kipf & Welling, 2016a), GAT (Veličković et al., 2018), as well as SGC (Wu et al., 2019), which is a simplified GCN. For ResTran, we apply both non-NN vector-based models and NN-based models. For non-NN models, we apply label propagation (LP) (Zhou et al., 2003) and SVM (Cortes & Vapnik, 1995) with the Gaussian kernel for $X_G$. For NN models, we use two early and simple models, VAT (Miyato et al., 2018) and AVAE (Maaløe et al., 2016) to $X_G$. We only use FC layers and ReLU as an activation function for NN models, which are simple and established NN components. We ran our experiments on homophilous and heterophilous datasets. We conducted our experiments with the split where we know 5% labels, we use 25% for validation, and the rest for the test. We conducted our experiments on 10 random splittings and reported the average.

The results are summarized in Table 2 and 3. On homophilous datasets, we observe comparable performances among GNNs and ResTran + NN models. On heterophilous datasets, we observe the performance improvement from GNNs to ResTran, sometimes even with SVM. This means that our ResTran is more robust to homophily bias. This robustness is expected from the construction of ResTran since, unlike GNNs, $X_G$ preserves not only homophilous information but also heterophilous information as seen in Sec. 4.1.

## 7 Conclusion

We considered a vertex classification task on the graph-with-features setting, where we have a graph with associated features. While the modern approach to this task has been GNNs, we took an alternative approach to overcome the homophilous biases in GNNs. Our approach was to transform the feature vectors to incorporate the graph topology and apply standard learning methods to the transformed vectors. For this approach, we proposed a simple transformation of features, which we call *ResTran*. We established theoretical justifications for ResTran from resistance, $k$-means, and spectral clustering viewpoints. We also discusses of why ResTran is robust to homophilous biases. We empirically demonstrated that ResTran is more robust on the homophilous bias than existing GNN methods. Limitation and future work are that we are unsure how much ResTran has an expressive power, as done in (Xu et al., 2019). We conjecture that the expressive power of ResTran is less than the 2-WL test. Thus, we speculate that we need a different setup for triangle counting problems.

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

# A    NOTE ON KRYLOV SUBSPACE METHOD

This section breifly explains the Krylov subspace method and its advantages over some natural ideas.

## A.1    KRYLOV SUBSPACE METHOD

In this section, the Krylov subspace method is an established way to approximate the solution of the linear algebraic solutions. In this case, we consider to approximate $f(A)\mathbf{b}$ for the matrix $A \in \mathbb{R}^{n \times n}$ and for a vector $\mathbf{b} \in \mathbb{R}^n$.

The $r$-th Krylov subsupace $\mathcal{K}_r$ for the matrix $A \in \mathbb{R}^{n \times n}$ and for a vector $\mathbf{b} \in \mathbb{R}^n$ is defined as

$$\mathcal{K}_r(A, \mathbf{b}) := \mathrm{span}\{\mathbf{b}, A\mathbf{b}, A^2\mathbf{b}, \dots, A^{r-1}\mathbf{b}\}. \tag{20}$$

The Krylov subspace method approximates $f(A)\mathbf{b}$ into this Krylov subspace $\mathcal{K}_r(A, \mathbf{b})$. To obtain this approximation, the common way is Arnoldi process. The Arnoldi process at $i$-th iteration obtains $Q_i \in \mathbb{R}^{n \times i}$ and $H_i \in \mathbb{R}^{i \times i}$ as

$$AQ_i = Q_i H_i + h_{i+1,i}\mathbf{q}_{i+1}\mathbf{e}_i^\top, \text{ where } Q_i := [\mathbf{q}_1 \dots, \mathbf{q}_i], \mathbf{q}_1 := \mathbf{b}/\|\mathbf{b}\|_2^2. \tag{21}$$

Note that $Q_i$ has orthonormal columns and $H_i$ is upper Hessenberg matrix. Then, Krylov subspace based method approximates

$$f(A)\mathbf{b} \approx Q_r f(H_r) Q_r^\top \mathbf{b} = \|\mathbf{b}\|_2 Q_r f(H_r)\mathbf{e}_1. \tag{22}$$

This process overall takes $O(rm)$ time complexity. Typically, $r$ is chosen small, say $r < 100$. See Higham (2008) for more details.

## A.2    ADVANTAGES OF KRYLOV SUBSPACE METHOD

This section discusses the advantages of the Krylov subspace method over some natural ideas.

One natural idea to approximate $L_b^{-1/2}X$ is to approximate $L_b^{-1/2}$ using polynomial function. This technique is commonly used, even in the GNN research area, such as Kipf & Welling (2016a). For example, we first expand $L_b^{-1/2}$ as

$$L_b^{-1/2} = a_0 I + a_1 L_b + a_2 L_b^2 + \dots, \tag{23}$$

and then approximate in some order, say,

$$L_b^{-1/2} \approx a_0 I + a_1 L_b. \tag{24}$$

While this is straightforwardly understandable, the Krylov subspace method approximates $L_b^{-1/2}X$ better as follows. While this polynomial approximation only uses $L$ when approximation, the Krylov subspace method approximates $LX$ using both $L$ and $X$ as seen in Appendix A.1. Hence, the Krylov subspace approximates $L_b^{-1/2}$ using more information than a polynomial approximation.

The other natural idea is to reduce the dimension, such as principal component analysis (PCA). We consider to eigendecompose the graph Laplacian as

$$L = \Psi \Lambda \Psi^\top, \tag{25}$$

where $\Psi := (\boldsymbol{\psi}_1, \dots, \boldsymbol{\psi}_n)$ and $\Lambda := \mathrm{diag}(\lambda_1, \dots, \lambda_n)$, where $\boldsymbol{\psi}_i$ is the $i$-th eigenvector and $\lambda_i$ is the $i$-the eigenvalue. Then, we compose $\Lambda_{r'} := \mathrm{diag}(\lambda_1, \dots, \lambda_{r'}, 0, \dots, 0)$. The value $r'$ is again typically small compared to $n$. Then, we approximate $L^{+1/2}$ as

$$L^{+1/2} \approx \Psi \Lambda_r^{+1/2} \Psi^\top. \tag{26}$$

This approximation can be conducted much faster than obtaining naively $L^{+1/2}$.

While dimensional reduction is the standard way to make pseudoinverse faster, the Krylov subspace method provides a better approximation in the following sense. Firstly, as the polynomial approximation, the Krylov subspace approximates $L_b^{-1/2}X$ with more information. Secondly, as discussed in 4.1 and as seen in the experimental result as 6, ResTran also works for heterophilous datasets. However, from the construction of the eigendecomposition, the reduction cut down the high-frequency information corresponding to the heterophilous information. Therefore, the dimensional reduction throws away the information that ResTran is good at dealing with.

## B  MORE RELATED WORK

This section provides more detailed related work to the resistance transformation and its application to learning problems.

Our justification relies on the connection between spectral clustering, effective resistance and $k$-means. The spectral clustering using ratio and normalized cut has been extensively studied (Fiedler, 1975; Shi & Malik, 1997). The Laplacian coordinate and effective resistance are used for the various learning problem such as clustering Fouss et al. (2007); Saito & Herbster (2023); Yen et al. (2008; 2005) and online learning Herbster & Pontil (2006); Herbster et al. (2005). The connection between normalized cut and weighted kernel $k$-means has been developed, such as Bach & Jordan (2003); Dhillon et al. (2004); Saito (2022). The connection between ratio cut, effective resistance, and $k$-means are loosely studied Saerens et al. (2004); Zha et al. (2001). However, these studies do not give the "exact" connection between ratio cut and spectral clustering like Thm. 8. Also, the previous studies do not give the Resistance Transformation interpretation. We discuss more details in Appendix G.

Since our ResTran aims to address the graph with feature problem, one popular approach to this problem is GNN. The GNN is firstly proposed as a neural network applied to the graph structural data (Gori et al., 2005; Scarselli et al., 2008). The GCN (Kipf & Welling, 2016a) and GAT (Veličković et al., 2018) are established methods. The recent advancements include (Gasteiger et al., 2020; Hamilton et al., 2017; Pei et al., 2020; Xie et al., 2016) to name a few; see (Wu et al., 2020) for more comprehensive survey. The closest approach in the sense of formulation to our ResTran is SGC Wu et al. (2019). The SGC aims to simplify $\ell$ layers of GCN. The SGC is formulated as

$$\hat{\mathbf{y}} = \text{softmax}(\tilde{A}^\ell X^\top \Theta), \text{ where } \tilde{A} := (D+I)^{-1/2}(A+I)(D+I)^{-1/2}, \ \Theta := \Theta^{(1)} \dots \Theta^{(\ell)}, \tag{27}$$

where $\Theta^{(i)}$ is a $i$-th layer of a fully-connected (FC) layer. This approach is close to ours for the following reason. If we apply $\ell$ layers of FC to ours, and then this can be written as $\hat{\mathbf{y}} = \text{softmax}(X_G^\top \Theta) = \text{softmax}(L_b^{-1/2} X^\top \Theta)$. The SGC is close since, in this setting, the difference is $\tilde{A}$ and $L_b^{-1/2}$. However, our approach is not limited to this formulation, but we can apply any building blocks, especially, activate functions such as ReLU. There have been some follow-ups on this simple approach (Chen et al., 2020; Salha et al., 2019; 2021; Zhu & Koniusz, 2021). Another relevant approach is PinvGCN Alfke & Stoll (2021). For a dense graph aiming for faster GCN, PinvGCN reconstructs three graphs by heuristic approximation of $L^+$, runs GCN for each graph, and then combines the results. While these studies heuristically simplify the GCN in some similar manner, we provide a theoretical justification on Resistance Transformation in Sec. 4. Also, again our ResTran is not limited to simplfied GCN models. In addition to various models of GNNs, transformers using the eigenvectors of Laplacian as positional encoding are considered (Dwivedi et al., 2023; Wang et al., 2022). Also, Convolutional GNNs also exploit spectral properties such as Bruna et al. (2014); Henaff et al. (2015). The polynomial approximation strategy is a standard practice to obtain the spectra of graph Laplacian, such as (Defferrard et al., 2016; Kipf & Welling, 2016a). Moreover, Krylov subspace method is used for the better approximation for the convolutional GNNs Luan et al. (2019). However, these studies are on specific GNNs while ours can be applied to any vector based model.

Some common problems to GNN are reported: limited expressive power Xu et al. (2019) and over-squashing (Di Giovanni et al., 2023; Topping et al., 2021; Black et al., 2023). The most relevant problem to this study is the "low-frequency bias" of GNNs, where GNNs tend to learn only homophilous information (Chang et al., 2021; Du et al., 2022; Hoang & Maehara, 2019; Hoang et al., 2020; Zheng et al., 2022; Zhu et al., 2003; Luan et al., 2022; Platonov et al., 2023). This phenomenon gets worse if we stack the GNN layers, which is known as "over-smoothing" (Li et al., 2018; Oono & Suzuki, 2019). By construction, our Resistance Transformation are expected to represent not only homophilous information but also heterophilous information.

Since this work is related to semi-supervised learning problem, this section reviews the SSl studies in detail. The SSL over graph is extensively studied (Blum et al., 2004; Zhou et al., 2003; Zhu et al., 2003). Unlike GNNs, these only uses the graph topology. The Planetoid (Yang et al., 2016) is an SSL method which incorporates features and the topology at the same time, while the most of the GNN models are known to outperform Planetoid. The SSL models are also discussed for the vector dataset. The early models include SVM-based one (Joachims, 1999), and early NN models (Ranzato & Szummer, 2008; Weston et al., 2008). Also, we apply a kernel function to the vector to form a

graph and apply the graph-based SSL models. The one of the early established deep neural network based SSL method is variational autoencoder (VAE) (Kingma et al., 2014), which is simplified by the follow-up study called Auxiliary VAE (AVAE) (Maaløe et al., 2016). Since then, there have been various improvements including (Laine & Aila, 2017; Miyato et al., 2018; Yang et al., 2022). However, none of these aim to incorporate the graph and features. Instead, we can apply these methods to our $X_G$, unless the models are not designed to some specific tasks, e.g., images (Berthelot et al., 2019; Kurakin et al., 2020; Sohn et al., 2020; Zhang et al., 2021).

## C    MORE DETAILS OF THE COORDINATE AND EFFECTIVE RESISTANCE

This section discuss the detials of the Coordinate and effective resistance, introduced in Sec. 2.2.

A symmetric PSD matrix $M$ induces a semi-inner product as $\langle \mathbf{x}, \mathbf{y} \rangle_M := \mathbf{x}^\top M \mathbf{y}$, where $\top$ denotes transposition. This inner product induces a semi-norm, as

$$\|\mathbf{x}\|_M := \langle \mathbf{x}, \mathbf{x} \rangle_M = \|M^{1/2}\mathbf{x}\|_2. \tag{28}$$

The reproduced kernel associated with the above semi-inner product is $M^+$, where $+$ denotes the pseudoinverse. We define the coordinate spanning set

$$\mathcal{V}_{M,\langle \cdot,\cdot \rangle_M} := \{v_i := M^+ \mathbf{e}_i : i = 1, \ldots, n\} \tag{29}$$

and let $\mathcal{H}_{M,\langle \cdot,\cdot \rangle_M} := \mathrm{span}(\mathcal{V}_{M,\langle \cdot,\cdot \rangle_M})$. This $\mathcal{H}_{M,\langle \cdot,\cdot \rangle_M}$ is a *Hilbert space* induced by inner product $\langle \cdot, \cdot \rangle_M$.

The set $\mathcal{V}$ acts as "coordinates" for $\mathcal{H}$, that is, if $\mathbf{w} \in \mathcal{H}$ we have $w_i = \mathbf{e}_i^\top M^+ M \mathbf{w} = \langle \mathbf{e}_i, M^+ \mathbf{e}_i \rangle_M$. Note that the vectors $\{\mathbf{v}_1, \ldots, \mathbf{v}_n\}$ are not necessarily orthonormal. We also remark that this coordinate property is simply the reproducing kernel property for kernel $M^+$ (Aronszajn, 1950). If we measure this space over the plain dot product $\langle \cdot, \cdot \rangle_2$, the coordinate is instead

$$\mathcal{V}_{M,\langle \cdot,\cdot \rangle_2} := \{v_i := M^{+1/2}\mathbf{e}_i : i = 1, \ldots, n\}, \tag{30}$$

since $\|M^+ \mathbf{e}_i\|_M = \|M^{+1/2}\mathbf{e}_i\|_2$. In the main text, for brevity, we use $\mathcal{V}_M := \mathcal{V}_{M,\langle \cdot,\cdot \rangle_2}$ and $\mathcal{H}_M := \mathcal{H}_{M,\langle \cdot,\cdot \rangle_2}$. For more details, see Herbster & Pontil (2006).

As discussed in Sec. 2.2, this coordinate spanning set using graph Laplacian (Laplacian Coordinate) plays a role. For the definition of the coordinate, we obtain the Laplacian coordinate by putting $M = L$. Note that the graph Laplacian is symmetric PSD. Now, we see that using $\mathbf{v}_i'' \in \mathcal{V}_{L,\langle \cdot,\cdot \rangle_L}$ and $\mathbf{v}_i \in \mathcal{V}_L$, we have

$$r_G(i,j) = \|\mathbf{v}_i'' - \mathbf{v}_j''\|_L^2 = \|\mathbf{v}_i - \mathbf{v}_j\|_2^2. \tag{31}$$

This relationship is how the effective resistance and the Laplacian coordinate are related.

## D    PROOFS FOR SECTION 4.2

This section provides the proofs for the claims in Sec. 4.2.

### D.1    PRELIMINARY SETUPS

This section set ups some preliminary facts.

Without loss of generality, we can reorder $G$ as $G = G_1 \cup \ldots \cup G_K$ and $|G_1| \leq \ldots \leq |G_K|$. For the visual aid of $J_G$, we can write $J_G$ as

$$J_G = \begin{pmatrix} \begin{smallmatrix} 1 \cdots\cdots 1 \\ \ddots \\ 1 \cdots\cdots 1 \end{smallmatrix} & & \\ & \ddots & \\ & & \begin{smallmatrix} 1\cdots\cdots 1 \\ \ddots \\ 1 \cdots\cdots 1 \end{smallmatrix} \end{pmatrix}, \tag{32}$$

Let $\mathbf{1}_{G_j}$ is all one vector for $G_j$, i.e., $(\mathbf{1}_{G_j})_i = 1$ if $j \in V_{G_j}$ otherwise 0. Then we have

$$(\mathbf{1}_{G_1} \cdots \mathbf{1}_{G_K})(\mathbf{1}_{G_1} \cdots \mathbf{1}_{G_K})^\top = J_G. \tag{33}$$

We also introduce the bound of resistance by the eigenvalue as follows.

**Lemma 9** (Chandra et al. (1996)). *For any $i, j \in V$, we have*

$$r_G(i, j) \leq \frac{2}{\lambda_2} \tag{34}$$

**Lemma 10** (Herbster & Pontil (2006)).

$$\max_i \|L^{+1/2}\mathbf{e}_i\|_2^2 \leq \max_{i,j} r_G(i, j) \tag{35}$$

By combining these two lemmas, we obtain

$$\max_i \|L^{+1/2}\mathbf{e}_i\|_2^2 \leq \frac{2}{\lambda_2} \tag{36}$$

## D.2 PROOF FOR PROP. 3 AND COR. 4

We conduct eigendecomposition on $L$, and obtain eigenpairs as $(\lambda_k, \psi_k)$. We define a matrix $U$ and diagonal matrix $\Lambda$ as

$$\Psi := (\psi_1, \psi_2, \ldots, \psi_n), \Lambda_{kk} := \lambda_k. \tag{37}$$

We remark that the psuedoinverse of $\Lambda$ can be written as

$$\Lambda_{ii} = 0, \quad \text{for } i = 1, \ldots, K \tag{38}$$

$$\Lambda_{ii}^{+1/2} = 1/\lambda_i^{1/2}, \quad \text{for } i \geq K + 1. \tag{39}$$

Now we define an $n \times n$ matrix $\Lambda_b$ which has only one element, as

$$(\Lambda_b)_{ii} = 1/n_{G_j} b \quad \text{for } i \in V_{G_j} \tag{40}$$

We can then write as

$$\begin{aligned} L_b^{-1/2} &= \Psi\Lambda^{+1/2}\Psi^\top + \sqrt{b}J_G \\ &= \Psi\Lambda^+\Psi^\top + \Psi\Lambda_b^{+1/2}\Psi^\top \\ &= \Psi(\Lambda^{+1/2} + \Lambda_b^{+1/2})\Psi^\top. \end{aligned} \tag{41}$$

Thus, for $\ell > K$, the eigenvector associated with $\lambda_\ell^{-1/2}$ is $\psi_i$. From Eq. (40), for $\ell \leq K$ the eigenvalue associated with $\psi_\ell$ is $\sqrt{n_{G_\ell}b}$, where $|G_1| \leq \ldots \leq |G_\ell| \leq \ldots \leq |G_K|$. If $n_{G_1}b > \lambda_2^{-1}$, $n_{G_i}b$ is the largest $K$ eigenvalues. This concludes the proof for Prop. 3.

Eq. (41) yields the Cor. 4.

Finally, by generalizing the fact that the square root of the all one matrix can be written as $(\mathbf{1}\mathbf{1}^\top)^{1/2} = \mathbf{1}\mathbf{1}^\top/\sqrt{n}$, we have

$$
J_G^{1/2} = \begin{pmatrix}
\begin{matrix} 1/\sqrt{n_1} \cdots\cdots 1/\sqrt{n_1} \\ \vdots \ddots \vdots \\ 1/\sqrt{n_1} \cdots\cdots 1/\sqrt{n_1} \end{matrix} & & \\
& \ddots & \\
& & \begin{matrix} 1/\sqrt{n_K} \cdots\cdots 1/\sqrt{n_K} \\ \vdots \ddots \vdots \\ 1/\sqrt{n_K} \cdots\cdots 1/\sqrt{n_K} \end{matrix}
\end{pmatrix}
\tag{42}
$$

From the proof of Prop. 3, we immediately have the following corollary.

**Corollary 11.** *Let $(\lambda_\omega, \psi_\omega)$ be the $\omega$-th eigenpair of $L$. Suppose that a graph $G$ is connected. If $nb > \lambda_2^{-1}$, $i$-th eigenpair $(\lambda_i^+, \psi_i^+)$ of $L_b^{-1}$ are*

$$
(\lambda_i^+, \psi_i^+) = (\lambda_{n+1-i}^{-1}, \psi_{n+1-i}) \text{ for } i = 1, \ldots, n-1, \quad (nb, \mathbf{1}/\sqrt{n}) \text{ for } i = n
$$

### D.3 PROOF FOR PROPOSITION 5

Using Cor. 4, we obtain

$$
\|\mathbf{v}_i' - \mathbf{v}_j'\|_2^2 = \|(\mathbf{v}_i + b\mathbf{1}^\top\mathbf{1}\mathbf{e}_i) - (\mathbf{v}_j - b\mathbf{1}^\top\mathbf{1}\mathbf{e}_i)\|_2^2 = \|\mathbf{v}_i - \mathbf{v}_j\|_2^2
\tag{43}
$$

Using the fact of Eq. (2), we conclude the proof.

### D.4 PROOF FOR PROPOSITION 6

Without loss of generality, we write as

$$
r_G(i,j) = \left\| \begin{pmatrix} L_{G_s}^{+1/2} + \sqrt{bn_{G_s}}\mathbf{1}_{G_s} \\ 0 \\ 0 \end{pmatrix} \mathbf{e}_i - \begin{pmatrix} 0 \\ L_{G_t}^{+1/2} + b\sqrt{bn_{G_t}}\mathbf{1}_{G_t} \\ 0 \end{pmatrix} \mathbf{e}_j \right\|_2^2
\tag{44}
$$

$$
= \left\| \begin{pmatrix} \sqrt{bn_{G_s}}\mathbf{1}_{G_s} \\ \sqrt{bn_{G_t}}\mathbf{1}_{G_t} \\ 0 \end{pmatrix} (\mathbf{e}_i - \mathbf{e}_j) - \begin{pmatrix} L_{G_s}^{+1/2} \\ L_{G_t}^{+1/2} \\ 0 \end{pmatrix} (\mathbf{e}_j - \mathbf{e}_i) \right\|_2^2
\tag{45}
$$

$$
\geq \left( \left\| \begin{pmatrix} \sqrt{bn_{G_s}}\mathbf{1}_{G_s} \\ \sqrt{bn_{G_t}}\mathbf{1}_{G_t} \\ 0 \end{pmatrix} (\mathbf{e}_i - \mathbf{e}_j) \right\|_2 - \left\| \begin{pmatrix} L_{G_s}^{+1/2} \\ L_{G_t}^{+1/2} \\ 0 \end{pmatrix} (\mathbf{e}_j - \mathbf{e}_i) \right\|_2 \right)^2 \quad (\because \text{ Triangle Inequality})
\tag{46}
$$

$$
= \left( (bn_{G_s} + bn_{G_t})^{1/2} - \left\| \begin{pmatrix} L_{G_s}^{+1/2}(\mathbf{e}_{G_s})_i \\ L_{G_t}^{+1/2}(\mathbf{e}_{G_t})_j \end{pmatrix} \right\| \right)^2
\tag{47}
$$

$$
\tag{48}
$$

We now show that the first term is strictly larger than the second term. The first term is bounded as

$$
bn_{G_s} + bn_{G_t} \geq 2bn_{G_1},
\tag{49}
$$

and

$$
\left\| \begin{pmatrix} L_{G_s}^{+1/2}(\mathbf{e}_{G_s})_i \\ L_{G_t}^{+1/2}(\mathbf{e}_{G_t})_j \end{pmatrix} \right\| = (\|L_{G_s}^{+1/2}(\mathbf{e}_{G_s})_i\|_2^2 + \|L_{G_t}^{+1/2}(\mathbf{e}_{G_t})_j\|_2^2)^{1/2}
\tag{50}
$$

$$\leq (\max_{i,j} r_{G_s}(i,j) + \max_{i,j} r_{G_t}(i,j))^{1/2} \tag{51}$$

$$\leq (2/\lambda_{K+1} + 2/\lambda_{K+1})^{1/2} \tag{52}$$

$$= 2\lambda_{K+1}^{1/2} \tag{53}$$

Therefore, due to the assumption that $b > (1 + \sqrt{2})^2/n_{G_1}\lambda_{K+1}$, we have

$$(bn_{G_s} + bn_{G_t})^{1/2} \geq \left\| \begin{pmatrix} L_{G_s}^{+1/2}(\mathbf{e}_{G_s})_i \\ L_{G_t}^{+1/2}(\mathbf{e}_{G_t})_j \end{pmatrix} \right\| \tag{54}$$

We also have if $\min x \geq \max y \geq 0$, then

$$(x - y)^2 > (\min x - \max y)^2 \tag{55}$$

since $x - y > \min x - \max y > 0$. By using these relations, we obtain

$$r_G(i,j) \geq \left( (bn_{G_s} + bn_{G_t})^{1/2} - \left\| \begin{pmatrix} L_{G_s}^{+1/2}(\mathbf{e}_{G_s})_i \\ L_{G_t}^{+1/2}(\mathbf{e}_{G_t})_j \end{pmatrix} \right\| \right)^2 \tag{56}$$

$$\tag{57}$$

$$\geq \left( (2bn_{G_1})^{1/2} - \frac{2}{\lambda_{K+1}^{1/2}} \right)^2 \tag{58}$$

$$\geq \frac{2}{\lambda_{K+1}} \geq r_G(i,j) \tag{59}$$

## E    PROOF FOR PROPOSITION 7

We now start with the standard $k$-means objective function using the general norm $\| \cdot \|$ is defined as

$$\mathcal{J}(\{C_j\}_{j=1}^k) := \sum_{j \in [k]} \sum_{i \in C_j} \|\mathbf{x}_i - \mathbf{m}_j\|^2, \quad \mathbf{m}_j := \sum_{i \in C_j} \mathbf{x}_i/|C_j|. \tag{60}$$

For each cluster $C_j$ of Eq. (60), we further rewrite the objective function of $k$-means as

$$\sum_{i \in C_j} \|\mathbf{x}_i - \mathbf{m}_j\|^2, \quad \mathbf{m}_j := \sum_{i \in C_j} \mathbf{x}_i/|C_j| \tag{61}$$

$$= \sum_{i \in C_j} \left( \|\mathbf{x}_i\|^2 - 2\sum_{\ell \in C_j} \langle \mathbf{x}_i, \mathbf{x}_\ell \rangle/|C_j| + \sum_{\ell \in C_j} \|\mathbf{x}_\ell\|^2/|C_j| \right) \tag{62}$$

$$= \sum_{i \in C_j} \|\mathbf{x}_i\| - 2\sum_{i,\ell \in C_j} \langle \mathbf{x}_i, \mathbf{x}_\ell \rangle/|C_j| + \sum_{i,\ell \in C_j} \|\mathbf{x}_\ell\|^2/|C_j| \tag{63}$$

$$= \sum_{i,\ell \in C_j} \|\mathbf{x}_i\|/|C_j| - 2\sum_{i,\ell \in C_j} \langle \mathbf{x}_i, \mathbf{x}_\ell \rangle/|C_j| + \sum_{i,\ell \in C_j} \|\mathbf{x}_\ell\|^2/|C_j| \tag{64}$$

$$= \sum_{i,\ell \in C_j} \left( \|\mathbf{x}_i\| - 2\langle \mathbf{x}_i, \mathbf{x}_\ell \rangle + \|\mathbf{x}_\ell\|^2 \right)/|C_j| \tag{65}$$

$$= \sum_{i,\ell \in C_j} \|\mathbf{x}_i - \mathbf{x}_\ell\|^2/|C_j| \tag{66}$$

$$\tag{67}$$

Summing up over the all cluster, we can rewrite Eq. (60) as

$$\mathcal{J}(\{C_j\}_{j=1}^k) = \sum_{j \in [k]} \sum_{i,\ell \in C_j} \|\mathbf{x}_i - \mathbf{x}_\ell\|^2/|C_j|. \tag{68}$$

By replacing $\mathbf{x}_i$ and $\mathbf{x}_j$ to $\mathbf{v}_i'$ and $\mathbf{v}_j'$, we conclude the proof.

## F    PROOF FOR THEOREM 8

We now rewrite Eq. (13) as

$$J(\{V_j\}_{j=1}^k) = \sum_{i \in V_j, j} (\|\mathbf{v}_i'\|_2^2 - 2\langle \mathbf{v}_i', \mathbf{m}_j \rangle_2 + \|\mathbf{m}_j\|_2^2)$$

$$= \sum_{\mathbf{v}_i' \in V_j, j} \left( \langle \mathbf{v}_i', \mathbf{v}_i' \rangle_2 - 2 \left\langle \mathbf{v}_i', \sum_{\mathbf{v}_l' \in V_j} \frac{1}{|V_j|} \mathbf{v}_l' \right\rangle_2 + \left\langle \sum_{\mathbf{v}_l' \in V_j} \frac{1}{|V_j|} \mathbf{v}_l', \sum_{\mathbf{v}_r' \in V_j} \frac{1}{|V_j|} \mathbf{v}_r' \right\rangle_2 \right)$$

$$= \sum_{i \in V_j, j} \left( (L_b^{-1})_{ii} - 2 \sum_{l \in V_j} \frac{1}{|V_j|} (L_b^{-1})_{il} + \sum_{l, r \in V_j} \frac{1}{|V_j|^2} (L_b^{-1})_{lr} \right) \tag{69}$$

$$= \sum_{i \in V_j, j} (L_b^{-1})_{ii} - \sum_{r, l \in V_j, j} \frac{1}{|V_j|} (L_b^{-1})_{rl} \tag{70}$$

$$= \text{trace} L_b^{-1} - \text{trace} Z_R L_b^{-1} Z_R, \tag{71}$$

where $Z_R$ is an $n \times k$ matrix which serves as an indicator matrix, defined in Sec. 2.5. Thus, if we minimize Eq. (71) with respect to $Z_R$, we maximize the second term. Assuming $Z_R$ is discrete, $Z_R^\top Z_R = I$. If we relax $Z_R$ with this constraint, $\text{trace} Z_R L_b^{-1} Z_R$ becomes a problem to obtain top $k$ eigenvectors. From Prop. 3 and Cor. 11, the top $k$ eigenvectors of $L_b^{-1}$ are equivalent to the smallest $k$ eigenvectors of $L$. Similarly to Sec. 2.5 case, using Cor. 3, optimal solutions of $k$-means on $\mathcal{H}_{L_b}$ and spectral clustering is given as the same set of vectors, which is the $k$ smallest eigenvectors of $L$. This completes the proof.

## G    COMPARISON WITH THEOREM 8 AND WEIGHTED KERNEL $k$-MEANS

This section expands the explanation in the main body on the comparison between Thm. 8 and the previous weighted kernel $k$-means. We recall that Thm. 8 revisits the spectral connection between $k$-means and spectral clustering, extensively studied as we saw in Sec. 2.5. However, the previous connections is different than Thm. 8 in a number of sense.

### G.1    FORMAL STATEMENT OF PROP. 2

We provide a formal statement of 2 and proof for Prop. 2.

Before we provide a formal statement, we need to define a "relaxed" solution of $k$-means. We start with rewriting $J_\phi(\{C_j\}_{j=1}^k)$ as a trace maximization problem as follows.

$$\mathcal{J}_\phi(\{C_j\}_{j=1}^k) = \text{trace} WKW - \text{trace} Z_M^\top W^{1/2} K W^{1/2} Z_M, \tag{72}$$

where $W$ is a diagonal matrix whose $i$-th element is $w(\mathbf{x}_i)$, $K$ is a Gram matrix, and indicator matrix $Z_M := W^{1/2} Z (Z^\top W Z)^{-1/2}$. We note that $Z_M \in \mathbb{R}^{n \times k}$ and $Z_M Z_M^\top = I$. See (Dhillon et al., 2004; Saito, 2022) for the detail of this rewriting. Now, relaxing $Z_M$, we can obtain the *relaxed* solution of the weighted kernel $k$-means.

Now, we provide a formal statement of Prop. 2 as follows.

**Proposition 12** ((Dhillon et al., 2004)). *Consider a graph $a_{ij} = \langle \phi(\mathbf{x}_i), \phi(\mathbf{x}_j) \rangle$ and its degree $d_i$. We apply spectral clustering to this graph A. We substitute a weight $w(\mathbf{x}_i) = 1/d_i$ to the weighted kernel $k$-means $\mathcal{J}_\phi(\{V_j\}_{j=1}^k)$ Eq. (4). Then, if we relax $Z_M$ and $Z_R$ we obtain*

$$\min_{Z_M \in \mathbb{R}^{n \times k}} \mathcal{J}_\phi(\{V_j\}_{j=1}^k) = \min_{Z_R \in \mathbb{R}^{n \times k}} \text{kNCut}(\{V_i\}_{i=1}^k) \tag{73}$$

In Prop. 12, we formalize the "relaxed sense" in Prop. 2.

Prop. 12 is proven as follows. Firstly, the normalized cut can be rewritten as

$$\min \text{kNCut}(\{V_i\}_{i=1}^k) = \max_{Z_N} \{\text{trace}(Z_N^\top D^{-1/2} A D^{-1/2} Z_N) \text{ s.t. } Z_N^\top Z_N = I\}. \tag{74}$$

This follows since $L_N = I - D^{-1/2}AD^{-1/2}$.

To minimize Eq. (72) w.r.t. $Z_M$, we want to maximize the second term of Eq. (72). From the definition in Prop. 12, we taking $W = D^{-1}$ and $K = A$. Then, we see that minimizing objective function Eq. (72) is equivalent to the normalized graph cut objective function Eq. (74). By this we observe the connection between normalized spectral clustering and weighted kernel $k$-means. For more details, see (Dhillon et al., 2004; Saito, 2022).

### G.2 COMPARISON BETWEEN THM. 8 AND PROP. 12

We now discuss the Thm. 8 and the previous result Prop. 12 from Dhillon et al. (2004).

**Vector vs. Discrete.** The previous spectral connection is applied to vectors but not discrete graph data. Seeing Eq. (4), the weighted kernel $k$-means only applies to the vector data $X = (\mathbf{x}_1, \ldots, \mathbf{x}_n)$. We construct a graph $G$ whose adjacency matrix is a gram matrix, i.e., construct a graph whose weight is

$$a_{ij} = k_{ij} = \phi(\mathbf{x}_i)^\top \phi(\mathbf{x}_j), \tag{75}$$

where $K$ is a gram matrix as defined in Sec. 2.5. The weighted kernel $k$-means is equivalent to the normalized cut on this graph. Thus, this previous connection assumes for the vector data. On the other hand, our connection can be for a "given" graph data $G = (V, E)$, and thus we do not have to assume any vector data.

**Laplacian Coordinate Insights.** Ours offers the Laplacian coordinate insights; seeing the Eq. (13), if we use $\mathbf{v}_i'$ to represent $i$-th vertex and put this vector into the standard $k$-means objective function, this is equivalent to the spectral clustering. On the other hand, the weighted kernel $k$-means cannot be applied to this setting; the previous connection does not incorporate our connection Thm. 8. Two potential scenarios to reach Laplacian coordinate insights can be considered. One is a kernel mapping scenario. A naive application of the weighted $k$-means to the previous framework is to use $L^+$ as a kernel and $\langle \cdot, \cdot \rangle_L$ as an inner product. However, this Eq. (4) is not equivalent to the discrete spectral clustering. The other scenario is incorporating the weight to the standard setting. Recall that our insights come from the standard $k$-means. Thus, if we aim the standard $k$-means from the weighted kernel $k$-means, we compute

$$\mathcal{J}_\phi(\{C_j\}_{j=1}^k) = \sum_{j=1}^k \sum_{i \in C_j} w(\mathbf{x}_i) \|\phi(\mathbf{x}_i) - \mathbf{m}_{\phi,j}\|^2, \quad \mathbf{m}_{\phi,j} := \sum_{\ell \in C_j} w(\mathbf{x}_\ell)\phi(\mathbf{x}_\ell) / \sum_{\ell \in C_j} w(\mathbf{x}_\ell) \tag{76}$$

$$= \sum_{j=1}^k \sum_{i \in C_j} \|w^{1/2}(\mathbf{x}_i)\phi(\mathbf{x}_i) - w^{1/2}(\mathbf{x}_i)\mathbf{m}_{\phi,j}\|^2. \tag{77}$$

However, this transformation does not go anywhere close to the standard $k$-means. To conclude, the previous connection does not incorporate Thm. 8, and thus does not offer the Laplacian coordinate insights.

**Normalized Cut Only vs. Ratio Cut AND Normalized Cut.** Finally, we would like to point out that the previous connection can only be applied to normalized cut. The previous connection depends on Eq. (74), which only holds for the normalized cut. If we substitute $W = D^{-1}$ and $A = K$, Eq. (72) becomes the top $k$ eigenproblem of $D^{-1/2}AD^{-1/2}$. This eigenproblem is equivalent to Eq. (6). Therefore, the previous connection can only be applied to normalized cut. On the other hand, our connection does not depend on Eq. (6) but on Eq. (71). By Eq. (70) ours can connect to the ratio cut. Furthermore, Thm. 8 naturally generalizes to normalized cut. Let $\mathbf{v}_i'' := \sqrt{d_i}\mathbf{v}_i'$. Then, we define the objective function and expand in a similar manner in Sec. F as

$$\mathcal{J}_N(\{V_j\}_{j=1}^k) := \sum_{j=1}^k \sum_{i \in V_j} \|\mathbf{v}_i'' - \mathbf{m}_j\|_2^2, \quad \mathbf{m}_j := \sum_{i \in V_j} \mathbf{v}_i'' / |V_j|, \mathbf{v}_i' \in \mathcal{V}_{L_b} \tag{78}$$

$$= \text{trace}\, D^{1/2}L_b^{-1}D^{1/2} - \text{trace}\, Z_R D^{1/2}L_b^{-1}D^{1/2}Z_R. \tag{79}$$

Therefore, minimizing Eq. (78) subject to $Z_R^\top Z_R = I$ is equivalent to top $k$ eigenvector problem of $D^{1/2}L_b^{-1}D^{1/2}$. This is equivalent to the smallest $k$ eigenvectors of $D^{-1/2}LD^{-1/2}$, by which we show that Thm. 8 naturally generalizes the ratio cut to the normalized cut.

Table 4: Homophilous Dataset Summary.

|       | Cora | Citeseer | Pubmed | Photo | Computer |
|-------|------|----------|--------|-------|----------|
| $|V|$ | 2708 | 3327 | 19717 | 7650 | 13752 |
| $|E|$ | 5429 | 4732 | 44338 | 119081 | 245861 |
| Classes | 7 | 6 | 3 | 8 | 10 |
| Features | 1433 | 3703 | 500 | 745 | 767 |

Table 5: Heterophilous Dataset Summary.

|       | Texas | Cornell | Wisconsin | chameleon | squirrel | actor |
|-------|-------|---------|-----------|-----------|----------|-------|
| $|V|$ | 183 | 183 | 251 | 2277 | 5201 | 7600 |
| $|E|$ | 295 | 309 | 499 | 31421 | 198493 | 26752 |
| Classes | 5 | 5 | 5 | 5 | 5 | 5 |
| Features | 1703 | 1703 | 1703 | 2325 | 2089 | 932 |

## H  EXPERIMENTAL DETAILS

This section discusses the experimental details of the main body. For ResTran, we used $b = 1/(n\lambda_{K+1})$, that is the condition of Thm. 8. Also, we used the Krylov subspace dimension $r = 20$.

**Datasets.** For the homophilous dataset, we used the standard citation network benchmark; Cora (Mc-Callum et al., 2000), Citeceer (Sen et al., 2008), and Pubmed (Namata et al., 2012). We also used the two Amazom co-purchase graphs, photo and computer (McAuley et al., 2015). The homophilous dataset statistics are summarized in Table 4 For heterophilous dataset, we used used web data, Wisconsin, Cornell, and Texas, all of which are a part of WebKB (Craven et al., 1998). We also used the wikipedia dataset chameleon and squirrel (Rozemberczki et al., 2021), as well as actor (Pei et al., 2020). The heterophilous dataset statistics are summarized in Table 4. Note that the difference between homophilous datasets heterophilous datasets has been discussed in a variety of the literatures, such as (Luan et al., 2022; Platonov et al., 2023).

**Unsupervised Learning Setting.** For the feature only and ours, we computed the edge weight with a Gaussian kernel ($\kappa(\mathbf{x}_i, \mathbf{x}_j) = \exp(-\sigma\|\mathbf{x}_i - \mathbf{x}_j\|^2)$) for two vectors $\mathbf{x}_i, \mathbf{x}_j$. We used free parameter $\sigma \in \{10^{-2}, \ldots, 10^3\}$. To gain the sparsity, we further constructed a 100-NN graph from these gram matrices, which is a common technique. We compute the smallest $k$ eigenvectors of unnormalized Laplacian for all three graphs. Then, we apply the standard $k$-means to the smallest $k$ eigenvectors in order to obtain the clustering results. Since the $k$-means algorithm depends on the initial condition, we repeated it 10 times and reported the average and standard errors. For Fig. 1(a), we plot the second and third eigenvectors obtained by Matlab. Since Cora has many independent components, the second and the third eigenvectors are not necessarily to be like Fig. 1(a). However, observations will not change even if we take the other eigenvectors associated with eigenvalue 0.

**Semi-supervised Learning Setting.** For a fair comparison, we endeavored to use the same settings for ours and comparison as much as possible. We used non-normalized features. For non-NN based models, we again used a Gaussian Kernel and used free parameter $\sigma \in \{10^{-2}, \ldots, 10^3\}$, as done in the unsupervised learning setting. For NN based methods, we used 2 hidden layers for both of ours and our comparisons. For all of the settings, we used a dropout rate of 0.2. We train all models for 100 epochs using the Adam optimizer. For our ResTran, we applied various simple and established machine learning models to $X_G$. The model for non neural network, we used LP and SVM, as well as an established neural network semi-supervised models, AVAE and VAT. For AVAE, the first FC layer contains 256 hidden units, and the second FC layer contains 128 hidden units. For VAT, the first FC layer contains 1028 hidden units, and the second FC layer contains 512 hidden units. Also, each layer was activated by ReLU. Finally, we passed to the output layer. For AVAE, we used the embedding dimension as 30 and the dimension of the auxiliary variable as 30. We used batch size 128. We applied the learning rate of 0.01 to Adam for AVAE. For the comparison, apart from the setting above, we used the implementation and hyperparameters as implemented in the examples of PYTORCH-GEOMETRIC[1]. Finally, remark that for citation network benchmarks, although various studies use the public splittings in Yang et al. (2016), we avoided using these since overfitting to this specific splitting is reported (Shchur et al., 2018).

---

[1]https://github.com/pyg-team/pytorch_geometric/tree/master/examples

**Details of the Methods Used in the Semi-supervised Experiments.** We discuss some details of the methods we used for ResTran. Recall that our experiments only used simple and established methods for both our proposal and the comparison since we want to exclude the effects of sophistication as much as possible. For non-NN models, LP (Zhu et al., 2003) is one of the established model in SSL, as we saw in Appendix B. The SVM (Cortes & Vapnik, 1995) is also an established model, while SVM is a supervised learning model in general. However, in this context, we can interpret the SVM as an SSL method, since, even though we only use the indices corresponding the training set, i.e., $\{(X_G)_{\cdot i}\}_{i \in Tr}$, in ResTran Eq. (10), the transformation uses the whole $L$ and $X$ but not $\{y_i\}_{i \in Te}$. Remark that we only use the training set $\{\mathbf{x}_{G,i}\}_{i \in Tr}$ to form a gram matrix and therefore the gram matrix is the size $|Tr| \times |Tr|$ matrix. For NN models, as we discussed in Appendix B, AVAE (Kingma et al., 2014) is a simpler version of the SSL via VAE, which is the one of the earliest NN based SSL models. Also, VAT (Miyato et al., 2018) is the one early established NN based SSL model using generative adversarial network behind the scene.

**Other Details.** We did not report the computational time since it is slightly difficult to have an apple-to-apple comparison of the computational time. The reason is that ours can exploit the pre-computation of Krylov subspace method, while no such pre-computation can be applied to the comparison methods. Also, our computational time depends on learning algorithms and architectures. However, both of the ResTran and GNNs take the same complexity; the ResTran costs $O(rfm)$, and GNNs takes $O(fmt)$, where $r$ and $t$ are constant. Our experiments were conducted on Google Colab Pro+, Matlab, and Mac Studio with M1 Max Processor and 32GiB RAM. Our implementation can be found in the supplementary material. Due to the size limit of openreview, we only supply a part of the dataset. Regarding the implementation, we plan to publish our code in GitHub, an online codebase repository service, in the final version. For the implementation of the comparison methods, we used the examples of PYTORCH-GEOMETRIC codes as discussed above.

# I  LEARNING IN HILBERT SPACE OF GRAPH

This section discusses the learning over the Hilbert space discussed in Appendix C. The conventional learning frameworks assume that the features reside in the Euclidean space. However, in our setting, the features are associated with the vertices of the graph. Thus, we assume that the feature vectors reside not in the Euclidean space but in the space induced from the graph. This section sets up such a learning framework.

## I.1  ENERGY OVER THE HILBERT SPACE OF GRAPH

We consider to learn in the Hilbert space $\mathcal{H}_{L,\langle\cdot,\cdot\rangle_L}$, which we defined in Appendix C. This Hilbert space is the same as the space we can define the effective resistance, as we discussed.

We now consider to embed the feature $\mathbf{u} \in \mathbb{R}^n$ into this space by the mapping $\mathbb{R}^n \to \mathcal{H}_{L,\langle\cdot,\cdot\rangle_L}$

$$\mathbf{u}' = L^+\mathbf{u}. \tag{80}$$

We then define the energy in this Hilbert space for feature $S_{G,\mathcal{H}}(\mathbf{u})$ as

$$S_{G,\mathcal{H}}(\mathbf{u}) := \mathbf{u}'^\top L \mathbf{u}' = \mathbf{u}^\top L^+ \mathbf{u} = \|L^{+1/2}\mathbf{u}\|_2^2. \tag{81}$$

For $f$ features $U = (\mathbf{u}_1, \ldots, \mathbf{u}_f) \in \mathbb{R}^{n \times f}$, we define the *total energy* $S_{G,\mathcal{H}}(U)$ as

$$S_{G,\mathcal{H}}(U) := \sum_{i=1}^{f} S_{G,\mathcal{H}}(\mathbf{u}_i) = \|L^{+1/2}U\|_{\text{Fro}}^2. \tag{82}$$

We finally remark on the shape of the feature matrices. In this section, the feature matrix $U$ focuses on the features, $U = (\mathbf{u}_1, \ldots, \mathbf{u}_f)$ and $\mathbf{u}_i \in \mathbb{R}^n$, whereas the feature matrix we consider in the main body focuses on the vertices, i.e., $X = (\mathbf{x}_1, \ldots, \mathbf{x}_n)$ and $\mathbf{x}_i \in \mathbb{R}^f$. In a rough notation, $U = X^\top$.

## I.2  $k$-MEANS OVER THE HILBERT SPACE OF GRAPH

Before we discuss the $k$-means over this Hilbert space, we consider an "knockout" of $L^{+1/2}$, which appears in the total energy Eq. (82). Let $\bar{L}^{+1/2}$ be a knockout of a matrix $L^{+1/2}$ using the true labels

$\mathbf{y}$, defined as

$$\bar{L}^{+1/2} := \begin{cases} (L^{+1/2})_{ij} & \text{if } y_i = y_j \\ 0 & \text{otherwise.} \end{cases} \tag{83}$$

In the Hilbert space, we assume the following property.

**Assumption 1.** *In the Hilbert space, $\forall U \in \mathcal{H}_{L,\langle \cdot, \cdot \rangle_L}$ we can approximate as $L^{+1/2}U \approx \bar{L}^{+1/2}U$.*

This approximation assumption becomes exact in the following scenario. We consider a graph $G$ which is a union of $k$ graphs, and the labels are associated with each graph. In this scenario, the graph Laplacian for this graph is a block diagonal of the graph Laplacians of each graph, and henceforth the approximation assumption becomes exact. Thus, the assumption makes sense if we assume that two vertices are in different clusters.

We now define a $k$-means function over this Hilbert space. We recall that $X^\top - Z_R Z_R^\top X^\top$ is a difference between the data points $X$ and mean centers $Z_R Z_R^\top X^\top$. We define a $k$-means energy function by measure this difference by the graph norm, i.e.,

$$\mathcal{J}_G''(\{V_i\}_{i=1}^k) := \mathcal{S}_G(X^\top - Z_R Z_R^\top X^\top) = \|X^\top - Z_R Z_R^\top X^\top\|_{L^+,\text{Fro}}^2. \tag{84}$$

As discussed in Sec. 2.5, if we measure by the standard Frobenius norm $\|\cdot\|_{\text{Fro}}$, this becomes the standard $k$-means function. Under the Assumption 1, we can further approximate $\mathcal{J}_G$ Eq. (84) by $J_G$ Eq. (12).

**Theorem 13.** *For the clustering result $\{V_i\}_{i=1}^k$ induced from the true labels $\mathbf{y}$, if $\forall U \in \mathcal{M}_G^n$, $\|L^{+1/2}U^\top - \bar{L}^{+1/2}U^\top\|_{\text{Fro}} < \epsilon$, then $\|\mathcal{J}_G''\{V_i\}_{i=1}^k - \mathcal{J}_G(\{V_i\}_{i=1}^k)\|_{\text{Fro}} < 2\epsilon$.*

This theorem shows that for the true clusters $\mathcal{J}_G(\{V_i\}_{i=1}^k)$ can be approximated by $J_G(\{V_i\}_{i=1}^k)$ with the twice of the error rate of the approximation of the Assumption 1. The approximated $k$-means in Eq. (12) can be seen as a change of the basis. We now observe that the $J_G$ in Eq.(12) is further rewrite with basis $\mathbf{v}' \in \mathcal{V}_{L_b}$ as

$$\mathcal{J}_G(\{V_i\}_{i=1}^k) = \sum_{j=1}^k \sum_{i \in V_j} \|\mathbf{v}_i'^\top X - \mathbf{m}_j'\|_2^2, \quad \mathbf{m}_j' := \sum_{i \in V_j} \mathbf{v}_i'^\top X / |V_j| \tag{85}$$

The standard $k$-means (Eq. (3)) can be rewritten as

$$\mathcal{J}(\{C_j\}_{j=1}^k) = \sum_{j=1}^k \sum_{\mathbf{x}_i \in C_j} \|\mathbf{e}_i^\top X - \mathbf{m}_j\|_2^2, \quad \mathbf{m} = \sum_{\ell \in C_j} \mathbf{e}_\ell^\top X / |C_j|. \tag{86}$$

Thus, comparing the approximated $k$-means over a graph Eq. (85) and the standard $k$-means Eq. (86), the approximation can be seen a change of the basis from $\mathbf{e}_i$ to $\mathbf{v}_i'$.

We also mention that if we consider the featureless setting $X = I$, by Thm. 8 the approximated $k$-means over a graph corresponds to the spectral clustering. Henceforth, Thm. 8 and Thm. 13 justifies our proposed method in Sec. 3 in the following two ways; i) the natural generalization of the standard $k$-means can be approximated by the Laplacian transformation (Thm. 13) ii) In the featureless setting where $X = I$, this approximation corresponds to the spectral clustering (Thm. 8).

## J  PROOF FOR THEOREM 13

This section gives the proof for Thm. 13.

### J.1  PRELIMINARY SETUP

We now start with the proof for the Lemma 16. This lemma holds not only for $L^{+1/2}$, but also the general knockout operation.

We consider a "knockout" operation for a general matrix $B$. Let $\bar{B}$ be a "knockout" of a matrix $B \in \mathbb{R}^{n \times n}$ using $\mathbf{y}$, defined as

$$\bar{B} := \begin{cases} b_{ij} & \text{if } y_i = y_j \\ 0 & \text{otherwise.} \end{cases} \tag{87}$$

Let $Y \in \mathbb{R}^{n \times k}$ be an one hot representation of $\mathbf{y}$. Then, we have a following property for this knockout.

**Proposition 14.** *Let $B$ and $\bar{B}$ be matrices constructed as Eq. (87) using the labels of $\mathbf{y}$. If $\|B - \bar{B}\|_{\mathrm{Fro}} < \epsilon$, then $\|BY(Y^\top Y)^{-1}Y^\top - \bar{B}Y(Y^\top Y)^{-1}Y^\top\|_{\mathrm{Fro}} < \epsilon$.*

**Corollary 15.** *Let $B$ and $\bar{B}$ be matrices constructed as Eq. (87) using the labels of $\mathbf{y}$, and $U \in \mathcal{H}^f_{L,\langle\cdot,\cdot\rangle_L}$ If $\|BU^\top - \bar{B}U^\top\|_{\mathrm{Fro}} < \epsilon'$, then $\|BY(Y^\top Y)^{-1}Y^\top U - \bar{B}Y(Y^\top Y)^{-1}Y^\top U\|_{\mathrm{Fro}} < \epsilon'$.*

Proofs for Prop. 14 and Cor. 15 are discussed in Sec. J.3 and Sec. J.4.

## J.2 Proof for Theorem 13

We start with lemma which immediately follows from Prop. 14 and Cor. 15. We first define $Y_R := Y(Y^\top Y)^{-1/2}$ as a counterpart of $Z_R$.

**Lemma 16.** *If $\|L^{+1/2}X^\top - \bar{L}^{+1/2}X^\top\|_{\mathrm{Fro}} < \epsilon$, then $\|L^{+1/2}Y_R Y_R^\top X^\top - Y_R Y_R^\top \bar{L}^{+1/2}X^\top\|_{\mathrm{Fro}} < \epsilon$.*

**Lemma 17.** *If $\|L^{+1/2}X^\top - \bar{L}^{+1/2}X^\top\|_{\mathrm{Fro}} < \epsilon$, then $\|Y_R Y_R^\top L^{+1/2}X^\top - Y_R Y_R^\top \bar{L}^{+1/2}X^\top\|_{\mathrm{Fro}} < \epsilon$.*

This lemma tells us that that $L^{+1/2}X^\top Y_R Y_R^\top$ can be approximated by $\bar{L}^{+1/2}Y_R Y_R^\top$ by the same error rate between $L^{+1/2}$ and $\bar{L}^{+1/2}$. Using this lemma and assumption, we consider to approximate the $k$-means over this Hilbert space as

$$\mathcal{S}_G(X^\top - Y(Y^\top Y)^{-1}YX^\top) \tag{88}$$

$$= \|X^\top - Y(Y^\top Y)^{-1}YX^\top\|_{L^+,\mathrm{Fro}} \tag{89}$$

$$= \|L^{+1/2}X^\top - L^{+1/2}Y(Y^\top Y)^{-1}YX^\top\|_{\mathrm{Fro}} \tag{90}$$

$$\approx \|\tilde{L}^{+1/2}X^\top - \tilde{L}^{+1/2}Y(Y^\top Y)^{-1}YX^\top\|_{\mathrm{Fro}} \tag{91}$$

$$= \|\tilde{L}^{+1/2}X^\top - Y(Y^\top Y)^{-1}Y\tilde{L}^{+1/2}X^\top\|_{\mathrm{Fro}} \tag{92}$$

$$= \|\tilde{L}^{+1/2} + \sqrt{b}J_G^{1/2}X^\top - Y(Y^\top Y)^{-1}Y(\tilde{L}^{+1/2} + \sqrt{b}J_G^{1/2})X^\top\|_{\mathrm{Fro}} \tag{93}$$

$$\approx \|L^{+1/2} + \sqrt{b}J_G^{+/2}X^\top - Y(Y^\top Y)^{-1}Y(L^{+1/2} + \sqrt{b}J_G^{1/2})X^\top\|_{\mathrm{Fro}} \tag{94}$$

$$= \|L_b^{-1/2}X^\top - Y(Y^\top Y)^{-1}YL_b^{-1/2}X^\top\|_{\mathrm{Fro}} \tag{95}$$

$$= \sum_{j=1}^k \sum_{i;y_i=j} \|\mathbf{v}_i'^\top X - \mathbf{m}_j'\|, \quad \mathbf{m}_j' := \sum_{i;y_i=j} \frac{\mathbf{v}_i'^\top X}{|y_i = j|} \tag{96}$$

In the approximations above, we use Assumption 1 and Lemmas. 16 and 17. The error factor of 2 occurs since we approximate the term twice, which is the proof of Thm. 13.

## J.3 Proof for Proposition 14

We start with the discussion of the condition that $\|B - \tilde{B}\|_{\mathrm{Fro}}$. From the condition, we can rewrite as

$$\|B - \bar{B}\|_{\mathrm{Fro}} = \left(\sum_{i \nsim j} b_{ij}^2\right)^{1/2} < \epsilon. \tag{97}$$

We now compute

$$(BY(Y^\top Y)^{-1}Y^\top - \bar{B}Y(Y^\top Y)^{-1}Y^\top)_{ij} = \begin{cases} \frac{1}{|V_\tau|}\mathbf{e}_j^\top \sum_{\ell \in V_\tau} B\mathbf{e}_\ell & \text{if } j \notin V_\tau \text{ where } i \in V_\tau. \\ 0 & \text{if } j \in V_\tau \text{ where } i \in V_\tau \end{cases} \tag{98}$$

Note that the element does not change even if we change $i$ within the same cluster. We then have

$$\left(\frac{1}{|V_\tau|}\mathbf{e}_j^\top \sum_{\ell \in V_\tau} B\mathbf{e}_\ell\right)^2 = \left(\frac{1}{|V_\tau|}\sum_{\ell \in V_\tau} b_{ij}\right)^2 \tag{99}$$

$$= \frac{1}{|V_\tau|^2}\left(\sum_{\ell \in V_\tau} b_{ij}\right)^2 \tag{100}$$

$$< \frac{1}{|V_\tau|}\sum_{\ell \in V_\tau} b_{ij}^2 \tag{101}$$

From the second line to third line, we use the following inequality;

$$\left(\sum_{i=1}^n a_i\right)^p < n^{p-1}\sum_{i=1}^n a_i^p. \tag{102}$$

From construction, we have the $|V_\tau|$ identical elements for all $i \in V_\tau$. Thus, we compute

$$\|BY(Y^\top Y)^{-1}Y^\top - \bar{B}Y(Y^\top Y)^{-1}Y^\top\|_{\text{Fro}} = \left(\sum_{ij}(BY(Y^\top Y)^{-1}Y^\top - \bar{B}Y(Y^\top Y)^{-1}Y^\top)_{ij}^2\right)^{1/2} \tag{103}$$

$$= \left(\sum_{i \not\sim j}\left(\frac{1}{|V_\tau|}\mathbf{e}_j^\top \sum_{\ell \in V_\tau; i \in V_\tau} B\mathbf{e}_\ell\right)^2\right)^{1/2} \tag{104}$$

$$< \left(\sum_{i \not\sim j}\frac{1}{|V_\tau|}\sum_{\ell \in V_\tau; i \in V_\tau} b_{\ell j}^2\right)^{1/2} \tag{105}$$

$$= \left(\sum_{i \not\sim j} b_{ij}^2\right)^{1/2} \tag{106}$$

$$= \epsilon. \tag{107}$$

From the second line to third line we use Eq. (101). From the third line to fourth line we use the fact that we have the $|V_\tau|$ identical elements for all $i \in V_\tau$. From the fourth line to fifth line we use Eq.(97).

## J.4   PROOF FOR COROLLARY 15

Similarly to Appendix J.3, from the condition, we can rewrite as

$$\|(B - \bar{B})U^\top\|_{\text{Fro}} = \left(\sum_{i,j}(B - \bar{B})_{\cdot i}U_{j\cdot}^\top\right)^{1/2} < \epsilon'. \tag{108}$$

We define as

$$b'_{ij} = (B - \bar{B})_{\cdot i}U_{j\cdot}^\top. \tag{109}$$

We now compute

$$((BY(Y^\top Y)^{-1}Y^\top - \bar{B}Y(Y^\top Y)^{-1}Y^\top)U^\top)_{ij} = \frac{1}{|V_\tau|}\sum_{\ell \in V_\tau}(B - \bar{B})_{\cdot \ell}U_{j\cdot}^\top \tag{110}$$

$$= \frac{1}{|V_\tau|}\sum_{\ell \in V_\tau} b'_{\ell j}, \tag{111}$$

where $i \in V_\tau$. The rest of the proof is same as Appendix J.3.

## K  SOCIETAL IMPACT

Lastly, we briefly remark on the societal impact. Since this is foundational work towards an alternative learning methods for graph with features and does not target any immediate application, we cannot foresee the shape of positive or negative societal impact which this work may have in future.

