# OpenReview forum: "ResTran: A GNN Alternative To Learn Graph With Features"
_ICLR.cc/2024/Conference — Submitted to ICLR 2024_

### Official Review · Reviewer_Fupb · 2023-10-28

**Soundness:** 1 poor
**Presentation:** 2 fair
**Contribution:** 1 poor
**Rating:** 3
**Confidence:** 5

**Summary:**

The authors propose an unsupervised method that first combine the node feature with graph topology into a node-wise embedding. Then apply any standard machine learning method for the downstream tasks.

**Strengths:**

-	The connection to some unsupervised graph learning method is interesting.

**Weaknesses:**

-	Insufficient literature survey on the related work.
-	The novelty is extremely limited. See comments below.
-	The claim that existing GNNs can only work well on heterophilic graphs with complicated architecture is false. The authors not only ignore the complete literature of spectral GNNs but also LINKX method.

My first concern with the work is that its literature survey on prior related works is insufficient. Note that one major claim of this paper is that current GNNs can not handle heterophilic graphs if not using complex architecture. However, this is apparent wrong as the line of spectral GNNs research tackles this problem with a very simple design [1,2,3]. Also, LINKX [4] is another simple architecture that has been shown superior performance on heterophilic graphs. Notably, spectral GNNs are shown to be capable of learning ``any’’ graph spectral filtering that is beyond just low-pass (homophily) and high-pass (heterophily) cases. It is surprising that the authors completely ignore this literature.

On the other hand, the idea of obtaining node embedding from node features and graph topology in an unsupervised fashion has also been proposed previously. One of the early model SIGN [5] propose to compute propagated features $X, AX, A^2X,\cdots,A^KX$ first (with $A$ being potentially normalized or use $L$ instead) and concatenate them as the node embedding for applying MLP in downstream tasks. This work has also led to a series of works focusing on scalable graph learning methods such as SAGN [6] and GAMLP [7] with similar ideas. The ResTrans method is just using the embedding $L^{-1/2}X$. Note that one potential drawback of ResTrans is its computational complexity. Indeed, as the authors mentioned, naively compute $L^{-1/2}$ is computationally infeasible. They propose to apply the Krylov subspace method, which essentially computes $X,LX,L^2X,\cdots,L^rX$ and is very similar to SIGN design. It is a surprise to me that the authors completely miss this line of work as well. Compared to these prior works, I think the novelty of the proposed method is relatively limited.

I would suggest the authors explain the difference of their method to at least SIGN and compare them carefully in the experiments. Also, I think the authors should also compare to some spectral GNN baselines such as those in [1,2,3]. Otherwise, it is hard to convince me that ResTrans is a good method for heterophilic graphs.

## References

[1] Adaptive Universal Generalized PageRank Graph Neural Network, Chien et al. ICLR 2021.

[2] Bernnet: Learning arbitrary graph spectral filters via bernstein approximation, He et al., NeurIPS 2021.

[3] How powerful are spectral graph neural networks, Wang et al. ICML 2022.

[4] Large scale learning on non-homophilous graphs: New benchmarks and strong simple methods, Lim et al., NeurIPS 2021.

[5] Sign: Scalable inception graph neural networks, Frasca et al. ICML GRL+ workshop 2020.

[6] Scalable and adaptive graph neural networks with self-label-enhanced training, Sun et al. 2021.

[7] Graph attention multi-layer perceptron, Zhang et al. KDD 2022.

**Questions:**

1.	Discuss and compare with spectral GNNs [1,2,3] in both methodologies for heterophilic graphs and experiments.
2.	Compare with SIGN methods in both methodology and experiments.

---

> ### Author Response · Authors · 2023-11-19
> **Thank you for the review!**
>
> Thank you for the review! We will address your concern on the existing methods.
>
> > Literature reviews
>
> Thank you for pointing out the literature. We first apologize that we should have been careful when we use "simple." We say simple since we separate the graph consumption and prediction; consumption is ResTran and we may use any vector based ML methods that include the existing ones such as LP, SVM, AVAE and so on. This may be phrased as "versatile." GNNs are designed to graph problems, while we can use any vector based methods, such as AVAE or SVM. In this sense, while we appreciate the work you raised [1,2,3,4], those are sophisticated GNN models, which are different from our aims.
>
> Thank you also for pointing out SIGN. The difference between SIGN and ResTran is as follows. SIGN learns the weight of $X, AX, \ldots, A^{r}X$ through a GNN framework, while we project $L^{+1/2}$ to Krylov subspace. In this sense, SIGN is a more established model. Also, since SIGN needs to learn the weight of  $X, AX, \ldots, A^{r}X$ and store them, SIGN needs space. In fact, $r$ for SIGN is typically smaller, like 5 or less as discussed in the SIGN paper. On the other hand, since our ResTran aims to $L^{+1/2}$, we don't have to store each weight of  $X, AX, \ldots, A^{r}X$, instead, we only need to have the summation of these. By this we have more space. Also, since the weight of $X, AX, \ldots, A^{r}X$ are learnt by GNNs, these have GNNs' advantages as well as ones' disadvantages, which we want to avoid.

---

> > ### Comment · Reviewer_Fupb · 2023-11-23
> >
> > Thank you for the comments. I keep my evaluation unchanged.

---

### Official Review · Reviewer_hGqh · 2023-11-01

**Soundness:** 2 fair
**Presentation:** 2 fair
**Contribution:** 2 fair
**Rating:** 5
**Confidence:** 2

**Summary:**

The paper seeks to addresses the vertex classification problem in a manner different to standard GNN research. The idea is to utilize standard spectral methods such as clustering and sparsification methods to define new embeddings. The authors experimentally demonstrate that these embeddings out-perform standard GNNs on some datasets. Specifically, the authors propose "Resistance Transformation" on feature vectors X: Simply transform X by utilizing the Laplacian basis used to compute effective resistances, and feed the resulting data to standard vectorial learning algorithms.

**Strengths:**

Devising novel positional encodings, to the extent of eliminating the need for message-passing mechanisms, might be a worthwhile idea to explore. The authors also place their method within the homoplily/heterophily narrative, arguing that their embedding has lesser homophilious bias as compared to standard GNNs.

**Weaknesses:**

I am not sure about the computational complexity of these methods: The authors should have included some experimental results on time complexity to indicate whether the usually expensive eigenvector computations can be justified instead of simple combinatorial message passing.

Spectral embeddings/methods have inherent limitations in the kind of data they can capture from a graph: They fail to capture relational aspects of data (such as node/edge-colors) and so on, unlike combinatorial message-passing algorithms. If the authors propose such a radical departure from standard GNN methods, they should investigate their method on a variety of datasets, such as molecular graphs or synthetic graphs arising from relational sources.

**Questions:**

1. The computational cost of spectral methods typically goes to O(n^3). Can the authors comment on the running time complexity of the Krylov-subspace based embedding and compare it with the run-time costs of a standard message-passing GNN? Have the authors carried out any experiments to compare the run-time costs of the two approaches? Especially, I would like to know the status on sparse graphs, where the pseudo-inversions might be way more costly than message-passing. And how does the time complexity scale with graph size? The empirical investigation considers only medium-sized graphs.

2. How does the proposed method differ from the commonly used positional encodings based on spectral properties of the input graph?

---

> ### Author Response · Authors · 2023-11-19
> **Thank you very much for the review!**
>
> Thank you very much for the review! We will address what you raised in weakness and question sections.
>
> > Complexity analysis
>
> We provided the complexity analysis on page 4. Basically, ResTran takes O(rfm), and r is typically small, e.g., our experiments we used 15. O(fm) is equivalent to the established GNN models.
>
> > Edge-coloring
>
> Thank you for the suggestion. As stated in future work, at this stage, we are unsure how much of the expressive power our ResTran has. We speculate that at least ResTran has expressive power as two established models, GCN and GAT. However, we leave these as future work.
>
> > Difference between ResTran and positional encoding
>
> As written in Appendix B, our ResTran can be plugged into any vector based model, while positional encoding using spectral features aims to graph transformer architectures.

---

> > ### Comment · Reviewer_hGqh · 2023-11-19
> >
> > Thanks for your answer. I intend to keep my original rating.

---

### Official Review · Reviewer_MjNb · 2023-11-01

**Soundness:** 2 fair
**Presentation:** 3 good
**Contribution:** 1 poor
**Rating:** 3
**Confidence:** 4

**Summary:**

In this paper, the authors proposed a simple architecture for node classification tasks in graph formatted dataset name ResTran. It utilized the well-known spectral clustering methods to first generate vector representation of nodes in graph that incorporate both node features and graph connectivity, then apply standard vector based ML methods to them for downstream task. ResTran was claimed to be robust to homophilous bias which is commonly seem in traditional GNN settings.

**Strengths:**

Pros:
- Extensive summary of spectral clustering and other preliminaries.
- Overall good written and easy to follow.
- Simplify structure.

**Weaknesses:**

Overall, I found the paper raise more concerns than it claimed to solve. Here are some of my major concerns:
- Not much novelty from traditional spectral clustering methods, most part of the papers are well-known results or naive extension of existing methods.
- Most of the background or preliminaries can be distilled into shorter context or put in appendix such as propositions from previous papers, it’s currently taking more than 2 pages of the main paper.
- No complexity analysis to support the claim that it’s less complicated than GNN.
- Using the shifted graph Laplacian term b to control the heterphilous information in feature map seems to require a lot of fine tuning. How to choose the hyperparameters (b, r, etc) in experiment section is not clear to me, based on the appendix the authors used a fixed value, some ablation study would be nice to see.
- There are multiple works in GNN that already support heterophilous dataset without over-smoothing. The authors’ claim about the lack of GNN is not valid. I would suggest the authors to at least do some comparison with the recent ones.
- Experiment section lack of comparison to more recent GNN works that also targeting at heterophilous datasets.

**Questions:**

It seems like the authors didn't include most recent works in heterophilous GNNs and most of the claims against GNNs are lack of support. To name a few, JKNet [Xu et al., 2018], H2GCN [Zhu et al., 2020a], Geom-GCN [Pei et al., 2020],  GPR-GNN [Chien et al., 2020], GPNN [Yang et al., 2022] and many more are all methods that work with heterophilous graph dataset. I would recommend the authors to at least go over the literature before making the final conclusion.

---

> ### Author Response · Authors · 2023-11-19
> **Thank you for the review!**
>
> Thank you.for the review. We will respond to what you raised in weakness and question sections.
>
> > Lack of Novelty of the methods.
>
> As summarized in Appendix G.2, our theorem 8 is novel for the following points. Mainly, while the previous studies (stated in Prop 2), such as (Dhillion et al., 2004), only hold for the vector data, Thm.8 holds for the "given" graph data. Note also that the proof of Thm.8 is NOT the off-the-shelf application of Prop 2. See more in Appendix G.2
>
> > Presentation of preliminaries
>
> Thank you for the suggestion. We will endeavor to further shorten the preliminaries. We also stress that our main claims are not incremental results of preliminaries.
>
> > No complexity analysis of ResTran.
>
> We provided the complexity analysis on page 4.
>
> > Parameters
>
> Thank you for pointing out. From our preliminary experiments, the choice of b does not affect much. Also, the choice of r does not affect either if r is larger than 10. Thus, for simplicity, we used the fixed b and r. We will further clarify this point.
>
> > Experiments
>
> We are aware that multiple GNN models address the homophily bias. However, in the experiments, as written in the manuscript, we wanted to exclude sophistication for both methods as much as possible. Thus, we did not compare with the advanced models you raised. At the same time, we do not use some complicated methods for ResTran either. See more for the reply to Fupb.

---

### Official Review · Reviewer_gotZ · 2023-11-10

**Soundness:** 2 fair
**Presentation:** 3 good
**Contribution:** 2 fair
**Rating:** 3
**Confidence:** 4

**Summary:**

This paper proposes ResTran as an alternative to GNNs that may not suffer from homophilous bias and over-smoothing. ResTran is to first transform node features using graph spectral information so that graph structural information can be preserved in the transformed features. After that, one can directly apply vector-based learning methods on the transformed features, e.g., SVM, for node classification tasks on graph-structured data. The authors justify ResTran theoretically by drawing connections from effective resistance, k-means, and spectral clustering and justify it empirically by comparing with three traditional GNN architectures over 11 datasets. The experiments show that ResTran can perform comparably with baselines on homophilous datasets and outperform them on heterophilous datasets.

**Strengths:**

1. ResTran may not be biased towards homophilous data like traditional GCNs.
2. The proposed feature transformation is simple and may already be effective in capturing topology information.

**Weaknesses:**

1. It seems to me ResTran is closely related to 1-layer message-passing neural networks (MPNNs), since it utilizes $L^+$ to transform node features $X$ and it is known that $ L^+_{ij} $ represents the effective resistance between two end nodes in the graph interpreted as an electrical network. While traditional MPNNs utilize adjacent matrix $A$ to transform $X$ and have weight 0 when there is no edge between two nodes, I find the key idea is similar, which sounds like ResTran is still in some sense a GNN.

2. I do not really see that ResTran is simpler than existing GNNs. Depending on the definition of complexity, I find the feature transformation is already non-trivial as it includes utilizing Krylov subspace method to approximate the transformed features. After that, from the experiments, it seems it still needs complex neural networks to get decent results, i.e., AVAE, and using simple methods such as SVM does not seem to work.

3. The authors claim that ResTran may not suffer from over-smoothing and can overcome homophilous bias, and I think these need to be further discussed. It appears it is because ResTran only utilizes $L^+$ transforms features **once** (somewhat like a 1-layer MPNN) that it does not suffer from those issues, but it may come at the cost of the capability of capturing topological information in graphs. More experiments need to be done to demonstrate its capability, for example, comparing ResTran with [1], where some simple tricks were proposed to improve GCNs and, even with different splits, it seems it significantly outperforms ResTran, especially on heterophilous datasets.

4. I am unsure if the comparison is fair. Different from traditional GNNs, which propagate features with $A$, use MLPs to make predictions, and train the network with a classification loss, it seems critical for ResTran to use some semi-supervised models such as VAT and AVAE to get good results. However, the node representations yielded by those GNNs are not trained in the same way, e.g., VAT involves adversarial training--it is known that adversarial training can also further improve GNNs' performance [2].


[1] Chen, Ming, et al. "Simple and deep graph convolutional networks." International conference on machine learning. PMLR, 2020.

[2] Kong, Kezhi, et al. "Robust optimization as data augmentation for large-scale graphs." Proceedings of the IEEE/CVF Conference on Computer Vision and Pattern Recognition. 2022.

**Questions:**

1. Are there any particular reasons for using a different dataset split instead of the one that has been widely used in the previous literature?
2. I was wondering how AVAE is used exactly for node classification tasks. Will ResTran + MLP + Cross-entropy work?

---

> ### Author Response · Authors · 2023-11-19
> **Thank you for the review!**
>
> Thank you for the review! We will respond to the weakness and questions
>
> > It seems to me ResTran is closely related to 1-layer message-passing neural networks (MPNNs).
>
> Our contribution is to propose ResTran. Using ResTran, we obtain a vector representation of the graph-with-features. We may apply the transformed vector to any vector-based ML method. From this sense, our aim can be explained as "separating" consumption of graph and prediction. Contrary, GNNs do these at the same time. If we use NN models, the first layer "touches" $L^{+1/2}$; in some sense, we may say this is a sort of GNN. However, again, the point is that we separate consumption of graphs and prediction, which is different from most the frameworks of GNNs.
>
> > I do not really see that ResTran is simpler than existing GNNs.
>
> We apologize for the confusion. We should have been careful when we use "simple." We say simple since we separate the graph consumption and prediction; consumption is ResTran, and we may use any ML methods that include the existing ones. To describe this separation, this may be phrased as, say, "versatile." GNNs are designed for graph problems, and therefore we apply GNNs only to some specific problems over graphs. On the other hand, we can use any vector-based methods, such as AVAE or SVM, which is more "versatile" methods than GNNs.
>
> > Comparison methods and Experimental settings
>
> We are aware that various GNNs address various problems. Also, there may be various vector-based methods that better work for ResTran. For example, there may be better NN layers for ResTran or some kernel functions that work better than Gaussian kernel together with SVMs. However, as written in the manuscript, in order to eliminate any sophistications for both GNNs and ResTran, we may want to keep our experimental setting as simple as possible. For this purpose, i) we use the most established models, and ii) We use these methods with the simplest settings. For GNNs, GCN and GAT are the two most established models. We used SGC since SGC is close to ResTran, as discussed in Appendix B. For ResTran, AVAE is a simpler version of VAE, and VAE is one of the most established models. VAT is one of the early established models for semi-supervised learning using NNs. Also, we use SVMs with the established kernel function, Gaussian. We use GNNs and ResTran with the simplest settings as written in the manuscript. Also, by this, we say our comparison is fair since we do not use any sophistication for both methods. We simply plug in three GNN methods and SVM, LP, NN based methods, all of which are already established models.
>
> > Are there any particular reasons for using a different dataset split instead of the one that has been widely used in the previous literature?
>
> As written in the manuscript, we avoided using public split since "overfitting" to for some classical datasets public split is reported (Shchur et al. 2018). Therefore, we use random splitting.
>
> > I was wondering how AVAE is used exactly for node classification tasks. Will ResTran + MLP + Cross-entropy work?
>
> Exactly! We consume vector representation by ResTran using MLP + Cross Entropy.

---

> > ### Comment · Reviewer_gotZ · 2023-11-23
> >
> > Thanks for your responses. I have also read other reviews, and I find that the concerns about method novelty and simplicity still remain. Thus, I will keep my initial score.

---

### Meta-Review · Area_Chair_vEfQ · 2023-12-15

**Metareview:**

This paper proposes ResTran, an unsupervised node representation learning method for heterophilic graphs. The idea shows certain promise, but several concerns have been raised in the reviews:


1. The literature review on related spectral GNN methods is insufficient. The novelty compared to existing unsupervised graph representation learning techniques needs more clarification.


2. There lacks complexity analysis and ablation studies to justify design choices (e.g. selection of hyperparameters).


3. The superior performance relies much on additional semi-supervised models like VAT and AVAE. Fair comparison to recent GNN methods is insufficient.

I would encourage the authors to consider a major revision, and the paper could be much stronger if these concerns could be properly addressed.

**Justification For Why Not Higher Score:**

N/A

**Justification For Why Not Lower Score:**

N/A

---

### Decision · Program_Chairs · 2024-01-16

Reject